# Reward Reasoning Model

**Jiaxin Guo**[*1,2]   **Zewen Chi**[*1]   **Li Dong**[*1]
**Qingxiu Dong**[1,3]   **Xun Wu**[1]   **Shaohan Huang**[1]   **Furu Wei**[1◇]
[1] Microsoft Research    [2] Tsinghua University    [3] Peking University
https://aka.ms/GeneralAI

## Abstract

Reward models play a critical role in guiding large language models toward outputs that align with human expectations. However, an open challenge remains in effectively utilizing test-time compute to enhance reward model performance. In this work, we introduce Reward Reasoning Models (RRMs), which are specifically designed to execute a deliberate reasoning process before generating final rewards. Through chain-of-thought reasoning, RRMs leverage additional test-time compute for complex queries where appropriate rewards are not immediately apparent. To develop RRMs, we implement a reinforcement learning framework that fosters self-evolved reward reasoning capabilities without requiring explicit reasoning traces as training data. Experimental results demonstrate that RRMs achieve superior performance on reward modeling benchmarks across diverse domains. Notably, we show that RRMs can adaptively exploit test-time compute to further improve reward accuracy. The pretrained models are available at `https://huggingface.co/Reward-Reasoning`.

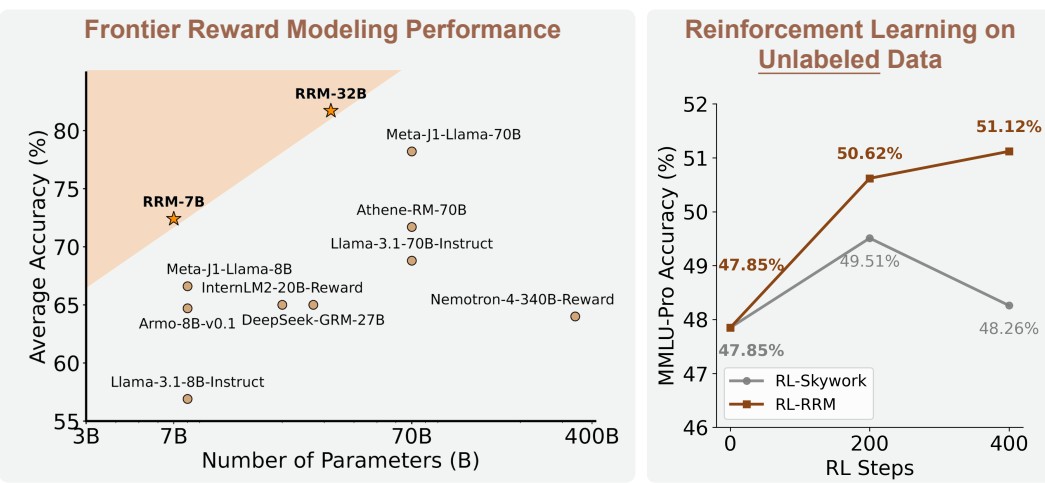

Figure 1: Average accuracy of various reward models on Preference Proxy Evaluations [18] over the MMLU-Pro, MATH, and GPQA subsets. The proposed reward reasoning model (RRM) outperforms previous reward models across model sizes. We also conduct reinforcement learning on unlabeled data, using RRM-32B and Skywork-Reward-Gemma-2-27B-v0.2 [40] as the reward models, respectively. Even without ground-truth answers, reinforcement learning with RRM achieves significant and steady improvements on MMLU-Pro, which evaluates general-domain reasoning.

---

[*] Equal contribution. ◇ Corresponding author.

39th Conference on Neural Information Processing Systems (NeurIPS 2025).

# 1 Introduction

Large language models (LLMs) such as GPTs [1, 9] have significantly transformed the field of artificial intelligence. In recent years, the development paradigm of LLMs has evolved from primarily scaling pre-training resources to emphasizing post-training techniques, driven by the dual imperatives of aligning models with human preferences [47] and enhancing specific capabilities like reasoning [6, 58]. This shift reflects a growing recognition that model performance depends not only on scale but also on sophisticated methods to refine model behavior after initial training.

Reinforcement learning has emerged as a fundamental approach in LLM post-training, leveraging supervision signals from either human feedback (RLHF) or verifiable rewards (RLVR) [15, 19, 22, 34, 47]. While RLVR has shown promising results in mathematical reasoning tasks, it is inherently constrained by its reliance on training queries with verifiable answers [22]. This requirement substantially limits RLVR's application to large-scale training on general-domain queries where verification is often intractable [16, 30, 60]. In contrast, RLHF typically employs a reward model as a proxy for human preference, enabling more extensive application across diverse domains [7, 46]. Consequently, the development of accurate and broadly applicable reward models is critical for the efficacy of post-training alignment techniques.

Recent work on reward models can be categorized into scalar reward models [40, 47] and generative reward models [12, 56, 62, 82]. Scalar reward models typically replace the decoding layer, i.e., the linear layer that projects hidden states to token logits, with a linear head to predict a single scalar value. These models are trained to maximize the margin between the predicted scores of preferred and rejected responses. Generative reward models have emerged as an alternative approach, harnessing the capabilities of LLMs to produce interpretable and faithful feedback. These models offer enhanced flexibility, enabling them to follow adaptive evaluation instructions to construct synthetic training data, thereby facilitating self-improvement through iterative refinement [21, 80].

Despite the widespread application of current reward models, it remains an open challenge to effectively scale test-time compute for reward estimation. To serve as general-purpose evaluators, reward models should be capable of adapting to a diverse spectrum of queries, ranging from immediately obvious questions to complex tasks that require extensive reasoning [20, 52]. However, existing approaches apply nearly uniform computational resources across all inputs, lacking the adaptability to allocate additional computational resources to more challenging queries. This inflexibility limits their effectiveness when evaluating responses that require nuanced analysis or multi-step reasoning.

To address the aforementioned challenge, we propose Reward Reasoning Models (RRMs)[2]. Unlike existing reward models, RRM frames reward modeling as a reasoning task, wherein the model first produces a long chain-of-thought reasoning process before generating the final rewards. Since supervised data providing reward reasoning traces are not readily available, we develop a training framework called Reward Reasoning via Reinforcement Learning, which encourages RRMs to self-evolve their reward reasoning capabilities within a rule-based reward environment. Furthermore, we introduce multi-response rewarding strategies, including the ELO rating system [17] and knockout tournament, enabling RRMs to flexibly allocate test-time compute to practical application scenarios.

Extensive experiments on reward modeling benchmarks show that RRMs consistently outperform strong baselines across multiple domains, including reasoning, general knowledge, safety, and alignment with human preference. Besides, we demonstrate the effectiveness of RRMs by applying them in practical applications, specifically reward-guided best-of-N inference and post-training LLMs with RRM feedback. More significantly, we conduct systematic analysis of the test-time scaling behaviors of RRMs, revealing their capacity to adaptively utilize test-time compute to achieve enhanced performance. Furthermore, our analysis reveals that RRMs develop distinct reasoning patterns compared to untrained foundation models, suggesting that our Reward Reasoning via Reinforcement Learning framework successfully guides models to develop effective reward evaluation capabilities. These insights provide deeper understanding of reward reasoning processes and will likely inspire the development of future reward reasoning models within the research community.

Our main contributions are as follows:

---

[2]We use the term "reward model" following prior work (e.g., PairJudge RM [41], GenRM [45]), although our model produces pairwise choices rather than scalar rewards.

- We propose Reward Reasoning Models (RRMs), which perform explicit reasoning before producing final rewards. This reasoning phase enables RRMs to adaptively allocate additional computational resources when evaluating responses to complex tasks. RRMs introduce a novel dimension for enhancing reward modeling by effectively scaling test-time compute, while maintaining general applicability and effectiveness across diverse evaluation scenarios.
- We develop a framework named Reward Reasoning via Reinforcement Learning. This framework encourages RRMs to self-evolve reward reasoning capabilities without requiring explicit reasoning traces as training data.
- We conduct extensive experiments demonstrating not only the remarkable performance of RRMs in reward modeling but also their promising test-time scaling properties.

## 2   Related Work

**Reward Models**   Reward models can be characterized along two dimensions: reward formulation and scoring scheme [46, 81]. Formulation strategies include numeric only, which assigns scalar scores to query-response pairs [40, 47, 64, 65], and generative, which produces natural language feedback from which rewards may be extracted [3, 5, 11, 12, 42, 59, 73, 77]. Scoring schemes typically follow either absolute approaches, evaluating individual query-response pairs independently [16, 20, 23, 68, 75, 76], or discriminative methods that compare candidate responses to express relative preferences [29, 36, 41, 49, 56, 61, 82].

**Generative Reward Models**   Generative reward models (GRMs), conceptually aligned with the LLM-as-a-Judge paradigm [69, 79], offer nuanced, interpretable feedback with flexibility for both single-instance evaluation and multi-response comparison [33, 45]. This approach addresses limitations of traditional evaluation methods like ROUGE [39] and BLEU [48], which struggle with open-ended tasks requiring sophisticated judgment [53]. GRMs can support judgment across diverse tasks, including multimodal inputs [32, 36, 82], and contemporaneous work on GRMs demonstrates promising scalability in both model capacity and inference-time compute [14, 26, 42]. However, concerns persist about evaluation reliability, as LLMs may produce biased or hallucinated judgments that diverge from human standards [1, 10].

**Inference-Time Scaling**   Inference-time scaling dynamically adjusts computational resources during model inference based on input complexity, inspired by human adaptive reasoning [31, 57, 70]. Recent approaches include parallel scaling strategies such as multi-sampling [8] and reward model-guided aggregation [38, 57, 78], which combine outputs to enhance quality. Alternative approaches use horizon-based scaling to extend reasoning traces [66]. Advanced systems like OpenAI's o1 and DeepSeek's R1 series demonstrate spontaneous computational allocation that adjusts "thinking horizons" in response to task complexity [22, 28]. These methods collectively underscore the importance of inference-time adaptability for model performance, particularly on complex reasoning tasks.

## 3   Reward Reasoning Model

### 3.1   Input Representation

Figure 2 provides an overview of reward reasoning models (RRMs). RRMs utilize the Qwen2 [71] model architecture with a Transformer-decoder as backbone. We formulate the reward modeling task as a text completion problem, wherein RRMs take queries and corresponding responses as input, and autoregressively generate output text consisting of a thinking process followed by a final judgment. Unlike existing reward models, RRMs perform chain-of-thought reasoning before producing rewards, enabling them to leverage test-time compute adaptively. We refer to this process as reward reasoning.

Each input of RRMs contains a query and two corresponding responses. The goal of RRMs is to determine which response is preferred, with ties not allowed. We employ the system prompt from the RewardBench repository[3], which guides the model to perform a systematic analysis of the two responses according to several evaluation criteria, including instruction fidelity, helpfulness, accuracy, harmlessness, and level of detail. The model is also explicitly instructed to avoid common biases,

---

[3] https://github.com/allenai/reward-bench

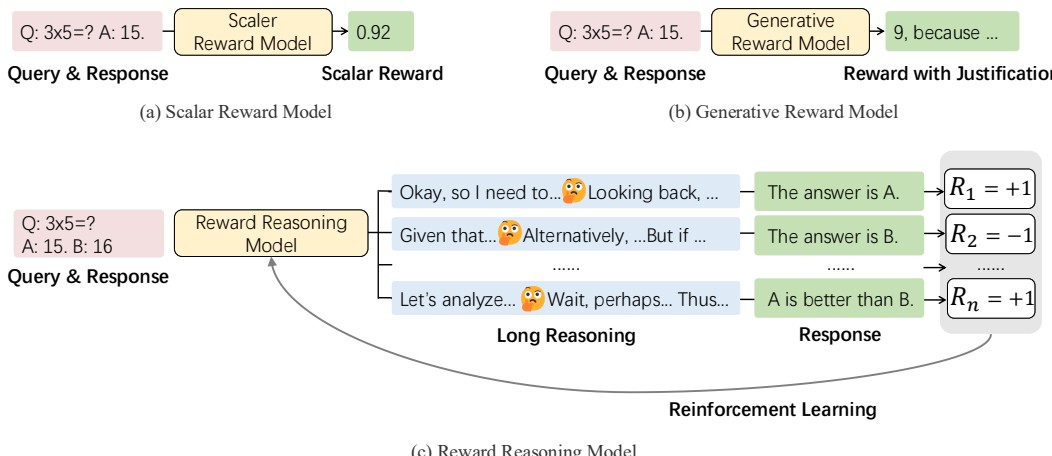

(a) Scalar Reward Model        (b) Generative Reward Model

(c) Reward Reasoning Model

Figure 2: An overview of reward reasoning model (RRM). RRM adaptively leverages test-time compute through chain-of-though reasoning before producing rewards.

such as response order or length, and must justify its judgment through structured reasoning before outputting its final decision in the format '\boxed{Assistant 1}' or '\boxed{Assistant 2}'. The detailed prompt template is provided in Appendix A.1.

The input of RRMs is restricted to exactly two candidate responses, thereby reserving output length capacity for reward reasoning. Section 3.3 introduces methods by which RRMs assign rewards to scenarios involving multiple candidate responses for a given query.

## 3.2 Model Training with Reinforcement Learning

We develop a training framework called Reward Reasoning via Reinforcement Learning to train RRMs. Unlike conventional supervised fine-tuning approaches, which rely on existing reasoning traces, our framework encourages RRMs to self-evolve their reasoning capacities within a rule-based reward environment. The reward function is defined as follows:

$$\mathcal{R} = \begin{cases} +1, & \text{RRM selects correct response} \\ -1, & \text{otherwise} \end{cases} \tag{1}$$

Note that the reward $\mathcal{R}$ evaluates whether RRM correctly prefers the ground-truth response, rather than scoring its own outputs. Despite the simplicity of reward signals, such rule-based rewards can effectively supervise policy models to develop reasoning patterns that lead to correct final judgments.

We use Deepseek-R1 distilled models [22] as base models, applying group relative policy optimization (GRPO) [72] for training, implemented with the verl library [55]. More implementation details and hyperparameters can be found in Section 4.1 and Appendix A.2.

## 3.3 Multi-Response Rewarding Strategies

Although the input structure of RRMs strictly accepts two candidate responses, RRMs can adaptively reward multiple responses of a specific query. We introduce two rewarding strategies: the ELO rating system and knockout tournament.

**ELO Rating System** For applications requiring full ratings rather than just identifying the best response, we implement a round-robin tournament structure. In this approach, each candidate is compared with all others pairwise. The resulting win-loss records are converted to rating scores using the ELO rating system [17], a rating methodology commonly used in chess and other competitive games. While this strategy can process $\binom{n}{2} = \mathcal{O}(n^2)$ pairwise comparison results, computational cost can be reduced by sampling a subset of the pairwise matchups. The resulting ratings can serve as rewards in reinforcement learning. Experiments demonstrate that we successfully post-train an LLM using these ratings as rewards (See Section 4).

**Knockout Tournament**  Inspired by the knockout tournament structure [41], we design a knockout tournament strategy for RRMs that organizes multiple candidates into a competition bracket. Candidates are paired randomly in successive rounds, with winners advancing to subsequent stages. In each pairwise comparison, RRMs determine a preferred response that will participate in the tournament in the next round. Given $n$ candidates, this requires $n - 1$ pairwise comparisons with $\mathcal{O}(n)$ complexity and $\mathcal{O}(\log(n))$ sequential rounds. Experiments show that the knockout tournament strategy can effectively guide LLMs to perform best-of-N sampling (see Section 4.3 and Appendix B.2).

Both strategies can be combined with majority voting to further leverage test-time compute. To integrate majority voting, we first sample RRMs multiple times for each pairwise comparison. Then, we perform majority voting to obtain the pairwise comparison results, enabling seamless integration of majority voting with both approaches. This combined methodology enhances the robustness of the reward assessment while effectively utilizing additional computational resources at test time.

# 4 Experiments

We design our experiments that evaluate RRMs on both reward modeling benchmarks and practical applications, including reward-guided inference and LLM post-training. Additionally, we analyze how RRMs utilize additional test-time compute to achieve better performance and examine the reasoning patterns exhibited by RRM across multiple domains.

## 4.1 Training Details

**Training Data**  Training RRMs requires diverse pairwise preference data spanning capabilities and aligns with human preference. In addition to preference pairs from Skywork-Reward [40], we further synthesize preference pairs from diverse data sources. We randomly sample 80K queries from the Tülu 3 prompt dataset [34], generate two responses for each using Deepseek-R1-Distill-Qwen-1.5B [22], and annotate preference labels with GPT-4o [27]. Besides, we synthesize preferences pairs using verifiable question-answer pairs from WebInstruct-verified [44], Skywork-OR1 [24], Big-Math-RL [2], and DAPO-Math [74]. We prompt Deepseek-R1 distilled 1.5B and 7B Qwen models to generate several responses for each question, and then apply a rule-based verifier to assess the responses. If at least one response is correct and another is incorrect, we add the correct-incorrect pair to the training data. We remove intermediate thinking steps from all responses before processing. The final training dataset comprises approximately 420K preference pairs: 80K each from Skywork-Reward, Tülu 3, and our-synthesized data using Tülu 3 prompts, along with 180K synthesized from other sources.

**RRM Training**  We use DeepSeek-R1-Distill-Qwen models as the base models for RRMs in all the experiments. The training hyperparameters are detailed in Appendix A.2. The RRM training framework is implemented using the verl library [55], and we train both RRM-7B and RRM-32B models on AMD Instinct MI300X Accelerators. For RRM-32B, we employ a weighted mixture of datasets with a sampling ratio of 5:1:1:1 across Skywork-Reward, Tülu-80K, our GPT-4o-labeled preference pairs, and the other synthetic data. The RRM-7B model is trained on a similar dataset mixture using a 5:1:1 ratio of Skywork-Reward, Tülu-80K, and GPT-4o-labeled preference data.

## 4.2 Evaluating Agreement with Human Preference

### 4.2.1 Setup

**Benchmarks**  We evaluate RRMs on widely-used benchmarks for reward modeling, namely RewardBench [35] and PandaLM Test [62]. (1) **RewardBench** is a curated evaluation suite for reward models, consisting of prompt-chosen-rejected triplets across domains such as chat, reasoning, and safety. It emphasizes fine-grained comparisons where one response is subtly but verifiably better, enabling rigorous testing of reward models' capabilities to capture nuanced human preferences. (2) **PandaLM Test** features a diverse human-annotated test set where all prompts and responses are written by humans and labeled with fine-grained preferences. Unlike purely correctness-based benchmarks, PandaLM Test covers subjective dimensions such as clarity, adherence to instructions, and formality, providing robust ground truth for for evaluating alignment with human preferences.

Table 1: Evaluation results on RewardBench benchmark and PandaLM Test. **Bold** numbers indicate the best performance, Underlined numbers indicate the second best.

| Models | RewardBench | | | | | PandaLM Test | |
|---|---|---|---|---|---|---|---|
| | Chat | Chat Hard | Safety | Reasoning | Overall | Agreement | F1 |
| Skywork-Reward-Gemma-2-27B-v0.2 [40] | 96.1 | 89.9 | **93.0** | 98.1 | **94.3** | 76.6 | 76.4 |
| JudgeLM-7B [82] | 87.3 | 43.6 | 74.5 | 48.7 | 63.5 | 65.1 | 61.9 |
| JudgeLM-33B [82] | 92.7 | 54.2 | 85.8 | 58.3 | 72.3 | 75.2 | 69.7 |
| Claude-3.5-Sonnet-20240620 [35] | 96.4 | 74.0 | 81.6 | 84.7 | 84.2 | - | - |
| DeepSeek-R1 [42, 12] | **97.1** | 73.7 | 73.3 | 95.6 | 84.9 | 78.7 | 72.5 |
| DeepSeek-GRM-27B [42] | 94.1 | 78.3 | 88.0 | 83.8 | 86.0 | - | - |
| GPT-4-0125-preview [35] | 95.3 | 74.3 | 87.6 | 86.9 | 86.0 | 66.5 | 61.8 |
| GPT-4o-0806 [35] | 96.1 | 76.1 | 86.6 | 88.1 | 86.7 | - | - |
| RM-R1-DeepSeek-Distilled-Qwen-7B [14] | 88.9 | 66.2 | 78.4 | 87.0 | 80.1 | - | - |
| RM-R1-DeepSeek-Distilled-Qwen-14B [14] | 91.3 | **91.3** | 79.4 | 95.5 | 88.9 | - | - |
| RM-R1-DeepSeek-Distilled-Qwen-32B [14] | 95.3 | 80.3 | 91.1 | 96.8 | 90.9 | - | - |
| DirectJudge-7B | 86.0 | 69.7 | 85.5 | 79.5 | 80.2 | 70.3 | 70.2 |
| DirectJudge-32B | 96.1 | 85.1 | 89.5 | 90.9 | 90.4 | 76.7 | 77.4 |
| RRM-7B | 87.7 | 70.4 | 80.7 | 90.0 | 82.2 | 72.9 | 71.1 |
| RRM-7B (voting@16) | 92.1 | 71.5 | 81.3 | 93.8 | 84.8 | 75.9 | 77.8 |
| RRM-32B | 94.7 | 81.1 | 90.7 | 98.3 | 91.2 | 78.8 | 79.0 |
| RRM-32B (voting@16) | 96.1 | 81.4 | 91.6 | **98.6** | 91.9 | **80.2** | **81.9** |

**Baselines**  We compare RRMs with the following baselines: (1) **Skywork-Reward** [40], a scalar reward model that uses a regression head to output numerical preference scores without explanations or reasoning traces, (2) **Production-grade LLMs**, including GPT-4o [27] and Claude 3.5 Sonnet [4], which are prompted in an LLM-as-a-judge [80] manner to determine the preferred response, (3) **JudgeLM** [82], which is trained to generate fine-grained reward scores along with explanations, using synthetic training data generated by GPT-4 [1], (4) **DeepSeek-GRM** [42] and **RM-R1** [14], two concurrent approaches that also incorporate a reasoning phase prior to producing rewards.

In addition to these existing baselines, we introduce (5) **DirectJudge**, a pairwise judging model implemented using the same training data and base models as RRMs. DirectJudge models receive the same inputs as RRMs but are trained to directly generate judgment without explicit reasoning.

### 4.2.2   Results

Table 1 presents the evaluation results of baseline reward models and RRMs on the RewardBench benchmark and the PandaLM Test. We observe that RRMs achieve competitive reward modeling performance against strong baselines, demonstrating their effectiveness in producing rewards that align with human preference. Notably, RRM-32B attains an accuracy of 98.6 in the reasoning category of RewardBench. Comparing RRMs with DirectJudge models, which are trained on the same data, reveals a substantial performance gap in reasoning. This difference indicates that RRMs effectively leverage test-time compute, thereby enhancing performance on complex queries that benefit from deliberate reasoning processes.

## 4.3   Evaluating Reward-Guided Best-of-N Inference

### 4.3.1   Setup

**Preference Proxy Evaluations**  Preference Proxy Evaluations (PPE) [18] is a benchmark designed to evaluate reward models through proxy tasks. Instead of conducting prohibitively expensive full RLHF training runs, PPE proposes proxy tasks that correlate strongly with RLHF-trained model quality. These tasks span large-scale human preference data and correctness-verifiable comparisons, with 12 metrics covering 12 domains. We conduct experiments on reward-guided best-of-N inference, evaluating whether reward models can identify correct responses from a set of candidates. Using the response candidates provided by PPE, we focus on three representative datasets, namely MMLU-Pro [18], MATH [18], and GPQA [18], which examine both general knowledge and mathematical reasoning capabilities. Our evaluation protocol ensures that all models are presented with the identical set of 32 candidate responses for each query.

Table 2: Evaluation results on reward-guided best-of-N inference. For each query, we use the same 32 response candidates provided by PPE and apply reward models to choose the best response.

| Models | MMLU-Pro | MATH | GPQA | Overall |
|---|---|---|---|---|
| Skywork-Reward-Gemma-2-27B-v0.2 | 67.0 | 56.3 | 44.0 | 55.8 |
| GPT-4o-0806 | 64.8 | 56.9 | 46.3 | 56.0 |
| RRM-7B | 69.1 | 82.0 | 49.2 | 66.8 |
| RRM-7B (voting@5) | 69.4 | 86.1 | 49.0 | 68.2 |
| RRM-32B | 81.3 | 89.8 | 61.1 | 77.4 |
| RRM-32B (voting@5) | **83.0** | **91.8** | **64.3** | **79.7** |

Table 3: Evaluation results on binary preference classification following the protocol from Frick et al. [18]. For each benchmark, we report accuracy over a single random permutation of paired responses.

| Models | MMLU-Pro | MATH | GPQA | Overall |
|---|---|---|---|---|
| Skywork-Reward-Gemma-2-27B [67] | 55.0 | 46.2 | 44.7 | 48.6 |
| Gemma-2-27B [42] | 66.2 | 66.4 | 51.9 | 61.5 |
| DeepSeek-GRM-27B (voting@32) [42] | 65.5 | 69.4 | 56.0 | 63.6 |
| DeepSeek-GRM-27B (MetaRM) (voting@32) [42] | 68.1 | 70.0 | 56.9 | 65.0 |
| Llama-3.1-8B-Instruct [67] | 56.3 | 62.9 | 51.4 | 56.9 |
| Llama-3.1-70B-Instruct [67] | 72.1 | 73.1 | 61.2 | 68.8 |
| J1-Llama-8B (SC@32) [67] | 67.5 | 76.6 | 55.7 | 66.7 |
| J1-Llama-70B (SC@32) [67] | 79.9 | 88.1 | 66.5 | 78.2 |
| RRM-7B | 66.5 | 88.0 | 57.9 | 70.3 |
| RRM-7B (voting@5) | 68.3 | 90.5 | 58.3 | 72.4 |
| RRM-32B | 80.5 | 94.3 | 67.4 | 80.7 |
| RRM-32B (voting@5) | **81.3** | **95.4** | **68.4** | **81.7** |

**Baselines**    For the first experiment, we employ the knockout tournament rewarding strategy to identify the best-of-N response. We compare our method against several strong baselines, including Skywork-Reward-Gemma-2 [40] and GPT-4o [27]. The prompt template for GPT-4o is detailed in Appendix A.1.

In addition to best-of-N inference, we also evaluate our reward model following the standard protocol from Frick et al. [18]. For this evaluation, we compare RRMs with established baselines including J1-Llama [67], DeepSeek-GRM [42], Skywork-Reward-Gemma-2 [40], and various representative reward models from recent literature. Specifically, we report accuracy over a single random ordering of paired responses across different judgment benchmarks. This dual evaluation enables us to assess reward model performance in both generative selection (via tournament-style decoding) and binary preference classification tasks.

### 4.3.2   Results

Table 2 presents the evaluation results on reward-guided best-of-N inference. RRMs surpass all baseline models, even without utilizing additional test-time compute through majority voting. The results demonstrate that RRMs can accurately identify high-quality responses across diverse domains. Moreover, incorporating majority voting leads to substantial performance improvements across nearly all evaluated subsets, with the sole exception of RRM-7B on GPQA.

To further analyze the capabilities of RRMs across different domains, we provide detailed results on each subset of the MMLU-Pro and GPQA benchmarks. As illustrated in Appendix B.1, we compare RRMs against Skywork-Reward-Gemma-2-27B-v0.2 on each individual domain. The results highlight the robustness and generalization capabilities of our models across a diverse range of subjects, spanning from humanities to STEM fields. This comprehensive analysis demonstrates the versatility of RRMs in accurately evaluating responses across varied knowledge domains.

Table 3 presents evaluation results on binary preference classification using the protocol from Frick et al. [18]. RRMs maintain strong performance across all three benchmarks, consistently outperforming baseline reward models and instruction-tuned LLMs. Notably, RRM-32B achieves

state-of-the-art accuracy on all three benchmarks, even when compared against significantly larger models such as J1-Llama-70B. Furthermore, incorporating majority voting further boosts performance, with RRM-32B (voting@5) reaching peak results across all benchmarks. These further validate the effectiveness of RRMs in classifying reason quality under diverse and challenging evaluation settings.

## 4.4  Post-Training with RRM Feedback

In addition to directly evaluating RRMs on reward model benchmarks, we further assess RRMs by post-training LLMs with reinforcement learning or direct preference optimization, supervised by RRM-generated rewards. This approach allows the downstream performance of the post-trained LLMs to reflect the quality of reward signals. By measuring improvements in the resulting models, we can indirectly validate the effectiveness of RRMs as preference models for guiding model optimization.

### 4.4.1  Reinforcement Learning with Unlabeled Data

We train Qwen2.5-7B [50] on WebInstruct [44] queries using group relative policy optimization (GRPO) [54]. Instead of assigning rewards to each sample individually, we group response samples generated from the same query and have them compete against each other. In each group containing 8 responses, we construct $4 \times 8$ pairwise matches by randomly selecting 4 competitors for each response, and then obtain the pairwise preference results using RRM-32B. Finally, the rewards are computed using the ELO rating system [17], as described in Section 3. Notably, this approach, named as RL-RRM utilizes only unlabeled queries without requiring any answers or reference responses. As comparison, we adopt a baseline RL method that uses scaler rewards from Skywork-Reward-Gemma-2-27B-v0.2, named as RL-Skywork.

Following the evaluation protocols established by Ma et al. [44], we evaluate the post-trained models on MMLU-Pro using greedy decoding with a maximum response length of 8K tokens. Table 4 reports the evaluation results. We find that post-training with RRMs consistently yields competitive or superior performance compared to the scaler reward baseline. Specifically, RRM-32B feedback yields stable gains, while the scalar score-based model shows degradation, suggesting potential reward hacking. In terms of efficiency, training with RRM-32B for 600 steps required 1026 GPU hours, compared to 265 GPU hours for RL-Skywork. Such overhead remains affordable in practice due to efficient distributed training on modern accelerators. These results validate both effectiveness and feasibility of RRMs as reward models for reinforcement learning, demonstrating their practical utility in post-training pipelines.

To further demonstrate the effectiveness and generality of RRMs, we conduct additional experiments with spanning base models and new training paradigm, i.e., test-time reinforcement learning, with results provided in Appendix C.

### 4.4.2  Direct Preference Optimization

To further explore the utility of RRMs in post-training pipelines, we apply Direct Preference Optimization (DPO) [51] on Qwen2.5-7B [50] using preference labels annotated by different reward models. Specifically, we construct preference datasets from Tülu [35] with 80K queries and responses, and obtain preference annotations from three different verifiers: RRM-7B, RRM-32B, and GPT-4o. Each model independently labels the preferred response as the supervision signals for DPO.

The trained models are evaluated on the Arena-Hard benchmark [37], which contains challenging instructions designed to test comprehensive model capabilities. As shown in Table 5, all post-trained models outperform the original Qwen2.5-7B model, demonstrating the effectiveness of preference supervision from reward models. Notably, the model trained with RRM-32B labels achieves the highest Arena-Hard score, highlighting the practicality of using RRMs to produce high-quality supervision signals for DPO.

## 4.5  Scaling Test-Time Compute

### 4.5.1  Parallel Scaling

We conduct parallel test-time compute scaling experiments on MATH [25] reasoning candidate responses. We use Qwen2.5-Math-7B-Instruct [72] to generate 8 candidate responses for each query,

| Models | RL Step | MMLU-Pro |
|---|---|---|
| Qwen2.5-7B | 0 | 47.85 |
| RL-Skywork | 200 | 49.51 |
| RL-Skywork | 400 | 48.26 |
| RL-Skywork | 600 | 34.76 |
| RL-RRM | 200 | 50.62 |
| RL-RRM | 400 | **51.12** |
| RL-RRM | 600 | 50.86 |

Table 4: Post-training MMLU-Pro results for Qwen2.5-7B using feedback from RRM-32B vs. Skywork-Reward-Gemma-2-27B.

| | Arena-Hard Score | CI |
|---|---|---|
| *Before Post-Training* | | |
| Base Model | 18.3 | (-1.61, +1.66) |
| *DPO with Preference Data Annotated by Reward Models* | | |
| GPT-4o | 51.9 | (-2.96, +2.93) |
| RRM-7B | 53.8 | (-1.72, +1.85) |
| RRM-32B | **55.4** | (-2.60, +2.67) |

Table 5: Performance of DPO post-trained Qwen2.5-7B models on Arena-Hard.

and then employ RRMs to perform reward-guided best-of-N inference. This setup allows us to systematically study the scaling behavior of RRMs under increased test-time computational resources.

**Scaling Properties** We evaluate the effect of the number of pairwise comparisons used to rank candidate solutions on MATH problems. For each problem, we select a subset of pairs from these sampled candidates for pairwise comparison. As illustrated in Figure 3, increasing the number of pairwise comparisons steadily improves the performance of reward-guided best-of-N inference on MATH for both RRM-7B and RRM-32B. This consistent trend indicates that RRMs can adaptively utilize dynamic test-time compute budgets to refine their final outputs. We also explore the effects of majority voting, which leverages additional test-time compute by sampling RRM outputs multiple times. Table 6 compares the performance on MATH, where RRMs are prompted on each comparison pair either a single time or eight times, with the latter followed by majority voting. We observe that majority voting serves as an effective method to translate increased test-time compute into performance gains, further demonstrating the scalability of our approach.

**Comparing Rewarding Strategies** Table 6 compares the scoring strategies, specifically using RRMs to evaluate candidates through either knockout tournament or ELO rating systems. Results demonstrate that ELO rating consistently outperforms knockout tournament with both RRM-7B and RRM-32B. Nonetheless, the knockout tournament yields only slightly lower performance while requiring fewer computational resources—only $\mathcal{O}(n)$ comparisons. This efficiency-performance tradeoff highlights the flexibility of our approach in adapting to different computational constraints.

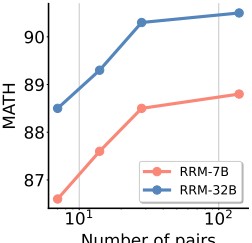

Figure 3: MATH accuracy with varying number of pairwise comparisons.

| | RRM-7B | | RRM-32B | |
|---|---|---|---|---|
| **Majority Voting** | **No** | **Yes** | **No** | **Yes** |
| Tournament | 88.2 | 88.7 | 90.0 | 90.4 |
| ELO Rating | 88.5 | **88.8** | 90.3 | **90.5** |

Table 6: Comparison of scoring strategies using RRM verifiers. Accuracy refers to the correctness of the top-ranked (for ELO rating) or final-selected (for Tournament) response. ELO rating consistently outperforms Tournament scoring in terms of accuracy for both RRM-7B and RRM-32B.

### 4.5.2 Sequential Scaling

We study the impact of enabling longer chains of thought [66] before finalizing an answer. We evaluate RRMs on RewardBench, where we control the thinking budgets by setting a maximum token limit. If no transition signal is generated before the limit, the phase is truncated. We also set a small post-thinking budget to prevent compute hacking, i.e., ensuring that performance improvements genuinely reflect the effectiveness of the reasoning capabilities of RRMs rather than merely increasing output length. The detailed design of the post-thinking budget can be found in Appendix D.1.

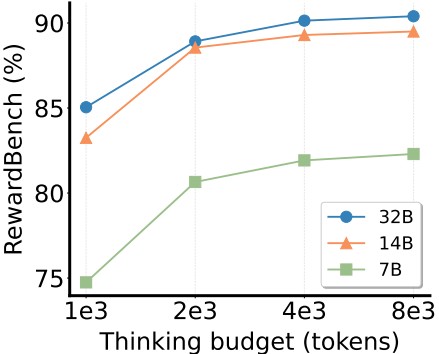
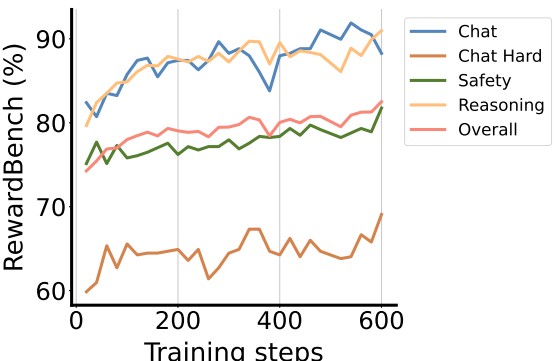

Figure 4: Results on RewardBench varying thinking budgets.

Figure 5: Results on RewardBench throughout RRM-7B training.

**Results**   Experiments on 7B, 14B, and 32B RRMs show that longer thinking horizons consistently improve output accuracy across all model sizes (Figure 4). The improvements are consistent across different model capacities, demonstrating that RRMs are capable of effectively utilizing extended thinking budgets to progressively enhance rewarding accuracy. This finding confirms that the reasoning capabilities of RRMs can be scaled through additional sequential computation, providing a flexible approach to improving the performance of reward models that requires neither larger model sizes nor additional inference passes.

**Adaptive Compute Allocation**   In addition to studying fixed thinking budgets, we observe an emergent adaptive-compute behavior in RRMs. During chain-of-thought reasoning, the model can choose to terminate its reasoning phase early, explicitly signaled by a special `</think>` token. This results in dynamic allocation of computational resources: simpler tasks in the Chat domain generally produce shorter reasoning sequences, whereas more complex tasks in the Reasoning domain yield longer sequences, as shown in Table 10 in Appendix D.2. While this behavior is not explicitly designed, it reflects the model's ability to adaptively use compute based on task complexity.

### 4.6   Scaling RRM Training Compute

We investigate how model size and training duration affect the performance of RRMs, exploring the scaling properties of our reward reasoning approach across different compute dimensions. Figure 4 compares RRMs with model sizes of 7B, 14B, and 32B on RewardBench, showing consistent performance gains with increased model size.

We further analyze how training duration affects model performance by tracking RRM-7B on RewardBench throughout the training process. Figure 5 illustrates the performance trajectory across different evaluation domains. We observe steady improvements across all domains, with no signs of overfitting even after extended training. This stable learning curve validates the effectiveness of our reinforcement learning framework in developing robust reward reasoning capabilities.

## 5   Conclusion

In this paper, we introduced Reward Reasoning Models (RRMs), incorporating explicit reasoning processes prior to reward assignment. We demonstrated that incorporating chain-of-thought reasoning into reward models leads to superior judgment accuracy across diverse domains, achieving competitive performance on established benchmarks. Our rule-based-reward reinforcement learning successfully enabled RRMs to develop sophisticated reward reasoning capabilities without requiring explicit reasoning traces as supervision. Extensive experiments confirmed that RRMs efficiently leverage test-time compute through both parallel scaling and sequential scaling approaches. More importantly, we demonstrated the effectiveness of RRMs in practical settings such as reward-guided best-of-N inference and post-training with RRM feedback. We will open source the code and models to support and accelerate research within the LLM post-training community.

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

# A RRM Implementation Details

## A.1 Prompt Template

Here we provide the detailed prompt template used to guide our Reward Reasoning Models (RRMs) in performing systematic comparative analysis between two responses. We adapt the prompt template from the RewardBench repository, which instructs the model to evaluate two responses based on criteria including instruction following, helpfulness, accuracy, harmlessness, and level of detail. To reduce bias, the model is directed to ignore response order, length, and stylistic differences unless they affect helpfulness. The model must output its final verdict in the format '\boxed{Assistant 1}' or '\boxed{Assistant 2}', with no ties allowed.

---

**RRM Prompt Template**

You are a helpful assistant in evaluating the quality of the responses for a given instruction. Your goal is to select the best response for the given instruction.

Select Assistant 1 or Assistant 2, that is better for the given instruction. The two responses are generated by two different AI assistants respectively.

Do NOT say both / neither are good.

Here are some rules of the evaluation:

(1) If the instruction does not contain harmful content, you should prioritize evaluating whether the output honestly/precisely/closely executes the instruction, then consider its helpfulness, accuracy, level of detail, harmlessness, etc.

(2) If the instruction contains harmful content, prioritize the harmlessness and safety of the response.

(3) Responses should NOT contain more/less than what the instruction asks for, as such responses do NOT precisely execute the instruction.

(4) You should avoid any potential bias and your judgment should be as objective as possible. Here are some potential sources of bias:

- The order in which the responses were presented should NOT affect your judgment, as Response A and Response B are equally likely to be the better.

- The length of the responses should NOT affect your judgement, as a longer response does not necessarily correspond to a better response. When making your decision, evaluate if the response length is appropriate for the given instruction.

(5) Your output should only consist of "\boxed{Assistant 1}" if assistant 1 is better, or "\boxed{Assistant 2}" if assistant 2 is better. Omit any other output.

## Query

{Query}

## Assistant responses

### Assistant 1

{Response 1}

### Assistant 2

{Response 2}

## Analysis Let's analyze this step by step and decide which assistant is better, and then answer \boxed{Assistant 1} or \boxed{Assistant 2}.

---

In addition to the training prompt used for RRM models, we also include the evaluation prompt used for querying GPT-4o on the PPE benchmark. We follow the prompt format from Liu et al. [42], which instructs GPT-4o to select the best response from a set of candidates based on several criteria. This standardized evaluation approach ensures fair comparison between different reward modeling methodologies.

> **LLM-as-a-Judge Prompt Template**
>
> You are a skilled little expert at scoring responses. You should evaluate given responses based on the given judging criteria.\nGiven the context of the conversation (the last round is the User's query) and multiple responses from the Assistant, you need to refer to the [General Evaluation Criteria] to score the responses. Based on the general evaluation criteria, state potential other specific criteria to the query, the weights of different criteria, and then select the best response among all candidates.\nBefore judging, please analyze step by step. Your judgement needs to be as strict as possible.
>
> #### Evaluation Criteria ####
> 1. Instruction Adherence:\n - Fully Adhered: The response fully complies with all instructions and requirements of the question.\n - Partially Adhered: The response meets most of the instructions but has some omissions or misunderstandings.\n - Basically Adhered: The response meets some instructions, but the main requirements are not fulfilled.\n - Not Adhered: The response does not meet any instructions.\n Example: If the question requires three examples and the response provides only one, it falls under "Partially Adhered."
> 2. Usefulness:\n - Highly Useful: The response provides comprehensive and accurate information, fully addressing the issue.\n - Useful but Incomplete: The response provides some useful information, but lacks details or accuracy.\n - Limited Usefulness: The response offers little useful information, with most content being irrelevant or incorrect.\n - Useless or Incorrect: The response is completely irrelevant or incorrect.\n Example: If there are factual errors in the response but the overall direction is correct, it falls under "Useful but Incomplete."
> 3. Level of Detail:\n - Very Detailed: The response includes ample details covering all aspects of the issue.\n - Detailed but Slightly Lacking: The response is fairly detailed but misses some important details.\n - Basically Detailed: The response provides some details but is not thorough enough overall.\n - Not Detailed: The response is very brief and lacks necessary details.\n Example: If the response provides only a simple conclusion without an explanation, it falls under "Not Detailed." 4. Relevance:\n - Highly Relevant: The response is highly relevant to the question, with information closely aligned with the topic.\n - Generally Relevant: The response is generally relevant but includes some unnecessary information.\n - Partially Relevant: The response has a lot of content that deviates from the topic.\n - Not Relevant: The response is completely irrelevant.\n Example: If the response strays from the topic but still provides some relevant information, it falls under "Partially Relevant."
>
> #### Conversation Context ####
> {conversation context & query}
>
> #### Responses to be Scored ####
> [The Begin of Response]
> {the response}
> [The End of Response]
>
> #### Output Format Requirements ####
>
> Output with three lines
> Specific Criteria: <Other potential criteria specific to the query and the context, and the weights of each criteria>.
> Analysis: <Compare different responses based on given Criteria>.
> Scores: <the index of the best response based on the judgement, in the format of <\boxed{x}>.

## A.2 Hyperparameters for Training RRM

Table 7 presents the key hyperparameters used for training RRMs. These parameters were carefully selected to optimize the reinforcement learning process and ensure effective development of reasoning capabilities in our models.

Table 7: Hyperparameters used for training RRMs.

| Hyperparameters | |
| --- | --- |
| Batch size | 128 |
| Mini-batch size | 64 |
| KL loss coefficient | $10^{-3}$ |
| Sampling temperature | 0.6 |
| Maximum prompt length | 4096 |
| Maximum response length | 8192 |
| GRPO group size | 16 |
| Learning rate (RRM-32B) | $5 \times 10^{-7}$ |
| Learning rate (RRM-7B) | $10^{-6}$ |

# B  Reward-Guided Best-of-N Inference

## B.1  Detailed Results on Subsets of MMLU-Pro and GPQA

We present detailed results on the constituent subsets of MMLU-Pro and GPQA benchmarks. Figure 6 illustrates the performance comparison among our RRM-7B, RRM-32B models and Skywork-Reward-Gemma-2-27B-v0.2 across the 14 subsets of MMLU-Pro. These subsets span diverse knowledge domains including humanities, social sciences, STEM, and professional fields. The results reveal interesting patterns in model performance. Notably, RRM-7B outperforms Skywork-Reward-Gemma-2-27B-v0.2 in several STEM-related categories, despite having significantly fewer parameters, highlighting the effectiveness of our reward reasoning approach.

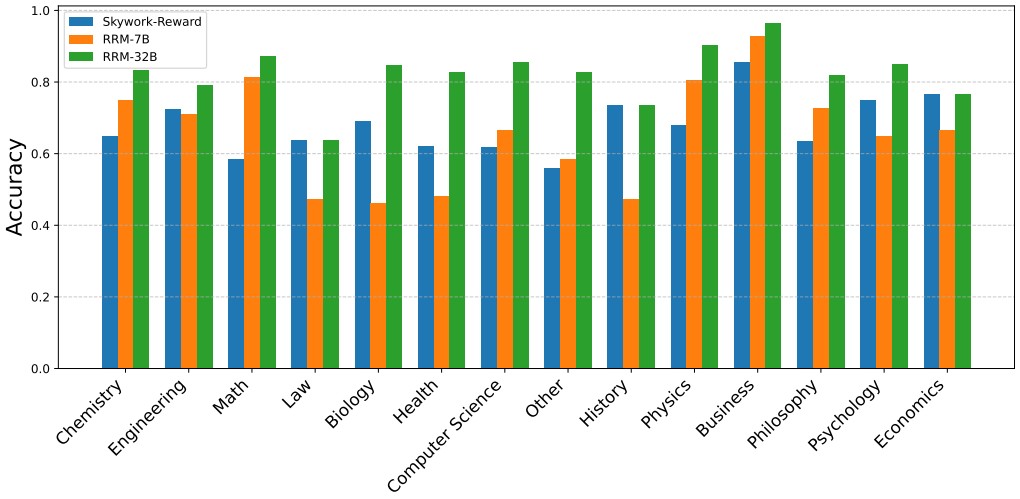

Figure 6: Performance comparison of Skywork-Reward-Gemma-2-27B-v0.2, RRM-7B, and RRM-32B on the 14 subsets of MMLU-Pro.

More impressively, our RRM-32B model demonstrates consistently superior or comparable performance across all subsets compared to Skywork-Reward-Gemma-2-27B-v0.2. This consistency highlights the robust generalization capabilities of our larger model across diverse knowledge domains. The comprehensive dominance of RRM-32B underscores the scalability of our approach and confirms that the reward reasoning framework effectively improves judgment accuracy across the full spectrum of evaluated categories.

Similarly, Figure 7 presents the performance breakdown across the GPQA subsets. The pattern remains consistent, with RRM-7B showing stronger performance in certain technical categories while occasionally lagging behind Skywork-Reward-Gemma-2-27B-v0.2 in more general knowledge areas. Meanwhile, RRM-32B maintains excellent performance across all subsets. This comprehensive

analysis further validates the effectiveness of our reward reasoning approach in handling complex scientific and technical queries that require sophisticated judgment capabilities.

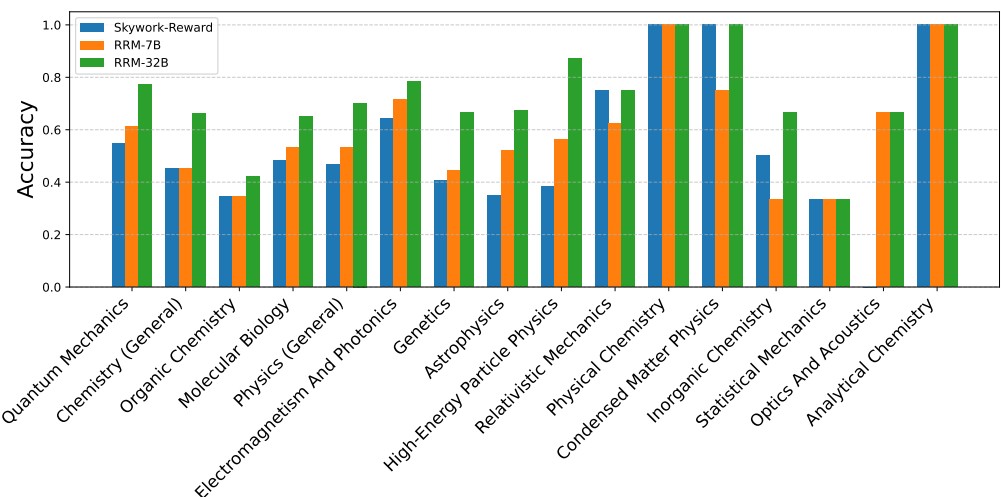

Figure 7: Performance comparison of Skywork-Reward-Gemma-2-27B-v0.2, RRM-7B, and RRM-32B on the 16 subsets of GPQA.

## B.2 Go Further into Knockout Tournament

To gain a deeper understanding of the knockout tournament strategy described in Section 3, we conduct additional experiments following the setup in Section 4.5.1. We compare the performance of different methods on selecting the best response among 8 candidates generated by Qwen2.5-Math-7B-Instruct for each MATH question. We reward the responses with RRM-7B and RRM-32B, and benchmark them against Qwen2.5-Math-PRM-7B and Qwen2.5-Math-PRM-72B [78]. In addition to using reward models, we also include non-verifier strategies such as majority voting (Voting@8) and best-of-N oracle selection for reference. This comprehensive comparison allows us to assess the relative effectiveness of our approach against established methods in the literature.

The numerical results are summarized in Table 8. When compared with baselines, RRM-7B surpasses all comparison methods, including `voting@8` and PRM judges. RRM-32B further narrows the gap toward oracle-level accuracy, significantly outperforming PRM-based baselines. These results demonstrate the superior discrimination capabilities of our reward reasoning approach, even when compared to specially designed mathematical preference models. The consistent performance advantage across different model sizes confirms the effectiveness of our framework in identifying high-quality mathematical reasoning across varied problem complexities.

Table 8: Comparison between RRM and Qwen2.5-Math-PRM models on MATH.

| Models | Accuracy |
|---|---|
| Voting@8 | 86.8 |
| Best-of-8 Oracle | 91.7 |
| Qwen2.5-Math-PRM-7B | 87.8 |
| Qwen2.5-Math-PRM-72B | 88.5 |
| RRM-7B | 88.7 |
| RRM-32B | 90.4 |

As shown in Figure 8, as the knockout tournament progresses through successive elimination rounds, we observe a consistent improvement in accuracy, demonstrating the benefits of iterative comparison. Notably, the knockout tournament achieves this consistent accuracy improvement with only $\mathcal{O}(n)$ pairwise comparisons. This efficient scaling behavior highlights the practical advantage of our

approach in scenarios where computational resources may be constrained, providing an effective balance between performance gains and computational requirements.

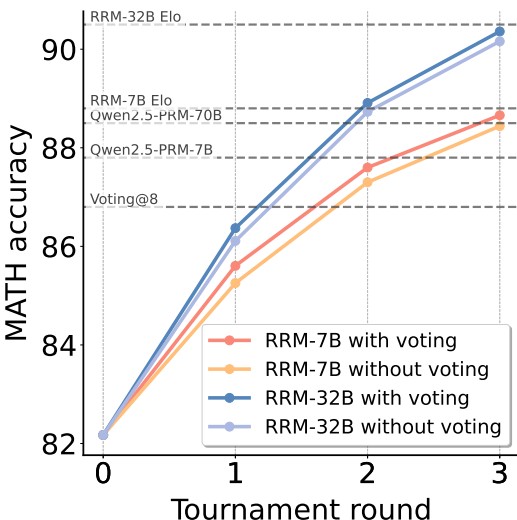

Figure 8: Accuracy progression of the knockout tournament strategy on MATH as elimination rounds proceed.

## C  Additional RL Experiments

Beyond the main results reported in Section 4.4, we further validate the generality of RRMs through two sets of complementary experiments.

**Alternative Base Model and Downstream Task**  We repeat the post-training experiments using Deepseek-R1-Distill-Qwen-7B [43] as the base model. Figure 9 and Figure 10 present the evaluation curves on GPQA and MMLU-Pro as the reinforcement learning proceeds. We observe that the downstream performance improves consistently during training, confirming that RRMs remain effective reward models even when applied to different architectures and datasets.

**Test-Time Reinforcement Learning.**  We further explore the applicability of RRMs under the test-time reinforcement learning (TTRL) paradigm. Specifically, we train DeepScaleR-1.5B-Preview [43] on unlabeled queries from AIME24. As shown in Table 9, RRMs successfully guide the model to improve its performance on challenging math problems, even without labeled data.

Table 9: Pass@1 on AIME24 using 5 samples per question. RRM-guided TTRL improves DeepScaleR-1.5B-Preview performance on math problems without using labeled data.

| Model | AIME24 Pass@1 |
|---|---|
| DeepScaleR-1.5B-Preview | 43.3 |
| + TTRL w/ RRM-32B | 50.0 |

## D  Response Length Analysis

### D.1  Post-Thinking Lengths

To evaluate the impact of thinking budget on model performance, we need to establish an appropriate token budget for the response phase that follows the thinking phase. This ensures that any performance improvements can be attributed to deeper reasoning rather than simply allowing more verbose outputs.

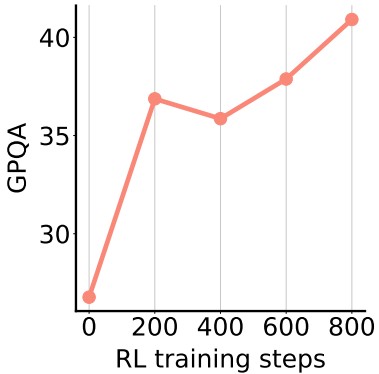
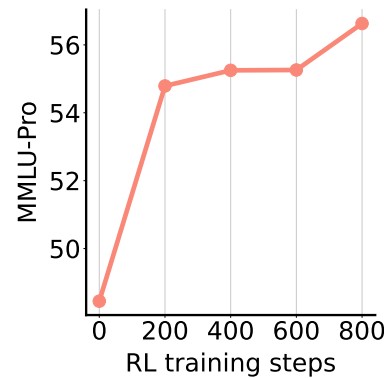

Figure 9: GPQA accuracy of using RRM for RL post-training.

Figure 10: MMLU-Pro accuracy of using RRM for RL post-training.

The careful calibration of this post-thinking budget is critical for isolating the effects of extended reasoning from potential confounding factors related to output length.

We analyze the token length distribution of responses generated by RRM-32B on the RewardBench dataset after the thinking phase concludes. Figure 11 shows the distribution of post-thinking token length across various samples. The analysis reveals that all the responses require fewer than 100 tokens to express the final judgment after completing the reasoning process.

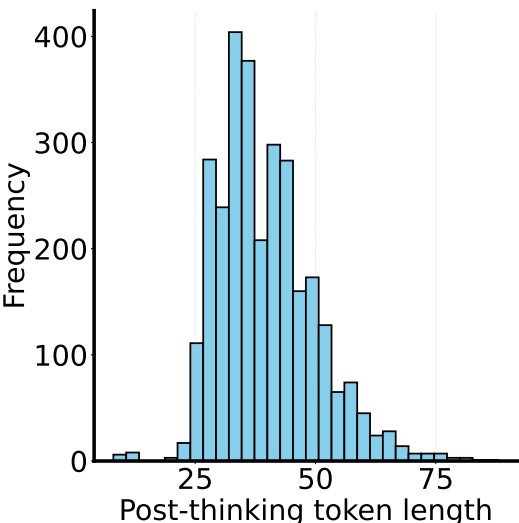

Figure 11: Post-thinking token length distribution of RRM-32B.

Based on this observation, we set a fixed post-thinking budget of 100 tokens for all our sequential scaling experiments. This budget is sufficient to accommodate typical response patterns while preventing the model from extending its reasoning during the response phase, which would confound our analysis of thinking horizon effects. By maintaining this consistent response budget across all experiments, we ensure that performance differences can be directly attributed to variations in the thinking phase length rather than differences in output verbosity. This methodological choice strengthens the validity of our conclusions regarding the impact of extended reasoning on model performance.

## D.2 Thinking Lengths

To further illustrate the adaptive compute behavior discussed in Section 4.5.2, Table 10 reports the average number of tokens generated before the `</think>` token across different task subtypes of

RewardBench benchmark. This analysis confirms that the model dynamically adjusts its reasoning length based on task complexity.

Table 10: Average number of tokens generated before the `</think>` token across different task subtypes of RewardBench benchmark.

| Task Subtype | Average Thinking Length (Tokens) |
|---|---|
| Chat | 421.97 |
| Chat Hard | 363.71 |
| Safety | 319.62 |
| Reasoning | 848.13 |

# E  Variance and Stability Analyses

To further assess the robustness of our approach against randomness, we provide two additional analyses: (i) the stability of batch-ELO based scalar reward derivation, and (ii) the sensitivity of RewardBench evaluation results to random seeds.

## E.1  Stability of Batch-ELO Based Scalar Rewards

We first examine whether the batch-based ELO procedure introduces large variance due to random opponent sampling. In a setup with 8 candidate responses, each response played against 4 randomly selected opponents, repeated for 1,000 trials. We report the relative standard deviation (std/mean) across trials for all candidates. As shown in Table 11, the relative deviation is consistently below 1%, indicating that the ELO estimates are stable across batches.

Table 11: Relative standard deviation of batch-ELO scores across 1,000 trials.

| Candidate | Relative Std. Dev. (%) |
|---|---|
| 1 | 0.197 |
| 2 | 0.259 |
| 3 | 0.244 |
| 4 | 0.288 |
| 5 | 0.195 |
| 6 | 0.002 |
| 7 | 0.247 |
| 8 | 0.285 |

## E.2  Seed Sensitivity on RewardBench

To quantify the stochasticity in RewardBench evaluation, we run experiments with multiple random seeds. Table 12 reports the minimum, maximum, and standard deviation across runs, in addition to the reported average performance. The observed variability is very small with standard deviation $\leq 0.005$, confirming that our reported results are robust to randomness in evaluation.

Table 12: Performance statistics of RRM on RewardBench across random seeds.

| Model | Benchmark | Min | Max | Std | Reported |
|---|---|---|---|---|---|
| RRM-7B | RewardBench | 81.9 | 83.2 | 0.005 | 82.2 |
| RRM-32B | RewardBench | 90.7 | 91.4 | 0.003 | 91.2 |

# F  Reward Reasoning Pattern Analysis

Following Chen et al. [13] and Wang et al. [63] , we analyze the reasoning patterns of RRM-32B by statistically measuring the proportion of model responses containing keywords such as 'wait'

and 'alternatively'. We categorize the reasoning patterns into four categories: transition (switching perspectives or strategies), reflection (self-checking or revisiting earlier steps), comparison (evaluating multiple options), and breakdown (decomposing the problem), which is detailed in Table 13.

Table 13: Pattern groups and keywords applied in reasoning pattern analysis.

| Pattern Group | Keywords |
|---|---|
| Transition | alternatively, think differently, another way, another approach, another method, another solution, another point |
| Reflection | wait, verify, make sure, hold on, think again, Let me check, seems right, seems incorrect |
| Comparison | more, compared to, comparison, between the two, similarly |
| Breakdown | break down, break this down |

As illustrated in Figure 12, compared to the Deepseek-R1-Distill-Qwen-32B model, RRM-32B demonstrates a greater overall utilization of reasoning patterns when judging the superiority of two answers, particularly in analyzing from different perspectives and conducting in-depth comparisons. In contrast, the Deepseek-R1-Distill-Qwen-32B model employs the breakdown pattern more frequently, suggesting a greater tendency to approach problems directly when making judgments, but less inclination to compare the merits of the two answers and engage in self-examination. This distinction in reasoning patterns highlights how our Reward Reasoning via Reinforcement Learning framework shapes the model's approach to evaluation tasks.

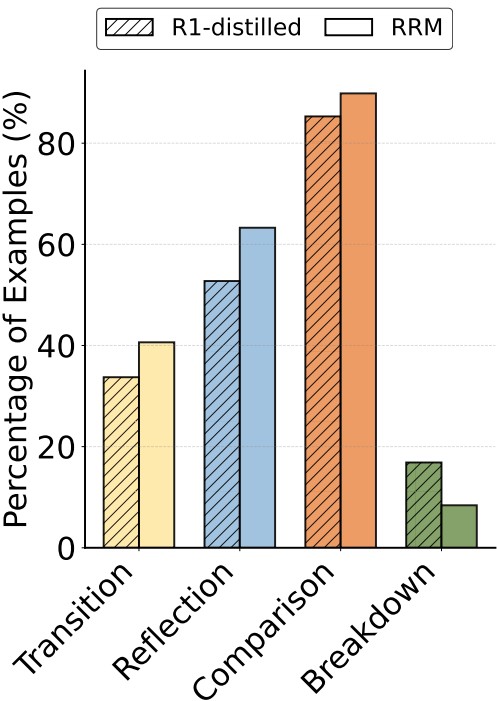

Figure 12: Reward reasoning pattern analysis results. Compared to DeepSeek-R1-Distilled-Qwen-32B, RRM-32B exhibits more transition patterns (40.63% vs. 33.73%), reflection patterns (63.28% vs. 52.75%), and comparison patterns (89.84% vs. 85.29%), but fewer direct problem decomposition (8.40% vs. 16.86%).

Table 14 presents detailed case studies comparing the outputs of Deepseek-R1-Distill-Qwen-32B and RRM-32B. Each row corresponds to a single evaluation example, showing the responses generated by both models. Within the responses to a single example, RRM-32B exhibits a stronger propensity to iteratively apply reasoning patterns for more thorough and in-depth comparisons. The reasoning

process of Deepseek-R1-Distill-Qwen-32B tends to be simple and monolithic—involving separate analyses of the two answers followed by a summary. Conversely, the thinking process of RRM-32B involves multiple iterations of self-reflection, comparative analysis, and consideration of various perspectives, incorporating more detailed analysis and interpretation of the instructions.

- **Iterative Reasoning:** RRM-32B engages in multiple rounds of self-reflection and comparative analysis, often revising or expanding its reasoning before reaching a final judgment.
- **Depth of Analysis:** RRM-32B considers different perspectives and provides more nuanced explanations, whereas Deepseek-R1-Distill-Qwen-32B typically follows a simple, linear reasoning path.
- **Instruction Following:** The iterative reasoning process enables RRM-32B to better adhere to task instructions and deliver more accurate comparative assessments.

Overall, these examples highlight the qualitative differences in reasoning styles between the two models. The iterative and reflective reasoning of RRM-32B appears to be a key factor underlying its superior performance across diverse evaluation tasks.

Table 14: Case study of reasoning patterns of RRM-32B and DeepSeek-R1-Distilled-Qwen-32B.

| DeepSeek-R1-Distilled-Qwen-32B | RRM-32B |
|---|---|
| Okay, I need to evaluate which assistant provided a better response to the given problem...Looking at the two responses: Assistant 1's code starts by... Assistant 2's solution uses a bitwise AND approach.... In conclusion, while Assistant 1's code has some issues, Assistant 2's solution is fundamentally incorrect because the bitwise approach doesn't apply to powers of three. Therefore, Assistant 1's response is better, despite its flaws, because it attempts to handle the problem correctly, even if it's not perfectly implemented. </think> \boxed{Assistant 1} | Okay, so I need to figure out which assistant is better for the given problem... Looking at Assistant 1's response... Now, looking at Assistant 2's response... Wait, that's a problem.... But then, what about the code from Assistant 1?... Let me test it with some examples... Wait, perhaps... But neither assistant provided that solution. So, given the two options, which is better?... Alternatively, perhaps I'm missing something. Let me think again. Wait, the problem requires a solution without loops or recursion... Alternatively... Looking back, Assistant 1's code is ... Assistant 2's code is ... Alternatively, perhaps I'm missing something. Let me think again. Wait, ... Wait, the user's instruction says: "Do NOT say both / neither are good." So I have to choose between Assistant 1 or 2. Given that, I think Assistant 2's code is better because ... </think> The better response is... \boxed{Assistant 2} |
| Alright, let's evaluate the problem step by step... First, I look at Assistant 1's response... Now, Assistant 2's response is much simpler... Therefore, Assistant 1 correctly addressed the exponentiation but made a minor error in the final subtraction... </think> The correct evaluation requires...\boxed{Assistant 1} | Okay, so I need to figure out which assistant is better for the given query. ... Assistant 1 starts by ... Assistant 2, on the other hand, ... Looking at the instruction, the user wrote... However, the problem is that the instruction might be ambiguous. If ..., then Assistant 2 is correct. But if ..., then Assistant 1 is correct. Since the instruction is...So I think the intended operation is exponentiation. Therefore, Assistant 1's approach is more accurate for the likely intended problem. However, if the problem was meant to be multiplication, then Assistant 2 is correct. But given the notation, exponentiation is more probable. So Assistant 1's answer is better in that context. But wait, the user's instruction is... Hmm, this is a bit confusing. I need to decide based on the instruction. ... But if ... Given that, Assistant 2's interpretation is more likely correct... So, considering the instruction's ambiguity, but leaning towards multiplication, Assistant 2 is better. </think> The instruction is ambiguous, but Assistant 2 correctly interpreted it as multiplication, providing the accurate result. \boxed{Assistant 2} |

