# OpenReview forum: "Reward Reasoning Models"
_NeurIPS.cc/2025/Conference — NeurIPS 2025 poster_

### Official Review · Reviewer_mL9a · 2025-06-08

**Clarity:** 2
**Significance:** 3
**Originality:** 3
**Rating:** 4
**Confidence:** 4

**Summary:**

The paper proposes reward reasoning models that can provide rewards to train models for better performance.
The LLM gets two candidate responses and it reasons about them (reasoning trace) and finally picks one of them. This is the "reasoning" in reward models. Furthermore, they consider training the model with RL to generate such a reward reasoning sequence. A major benefit of this is that even though RL is performed with binary rewards, the resulting reward model is able to generate non-binary intermediate feedback. They do extensive ablation behind the reasoning reward models towards alignment with human preference, best of N inference based on the reward model, fine-tuning of models with RRM, scaling compute budget during test-time, and effect of scaling RRM during training.

**Questions:**

What does it mean by adaptive compute? If it refers to increasing the hard budget of thinking (c.f. line 273), then it would suggest not using adaptive compute. The more or less use of the computing budget is not the result of the reasoning reward models. If there is a contribution of dynamically limiting the computation of RRM, then the adaptive compute terminology may be used.

Why didn't you consider any other models apart from Qwen?

**Ethical Concerns:**

["NO or VERY MINOR ethics concerns only"]

**Final Justification:**

I would like to keep my score of borderline accept. The idea of reasoning traces in reward models makes sense, and it is backed by good empirical performance. However, I would like to keep the borderline score since the paper does not discuss any limitations of the method throughout the paper. I have to explicitly ask to acknowledge the increase in computation overhead. Given this, I doubt the credibility of the results and hence kept it borderline.

**Limitations:**

No, limitations not addressed. I think authors should compare the computational complexity of the proposed reward model with other reward models, e.g. scaler. Since RRMs involve generating chain-of-thought traces and aggregating over multiple samples (e.g., voting @32), the inference cost is significantly higher.
Also, the paper never mentions any other limitations. I encourage the authors to be upfront and mention all of them during the rebuttal.

**Quality:**

2

**Strengths And Weaknesses:**

Strengths:
- The proposed idea of using reasoning traces in reward models makes sense, and itis  backed by good empirical performance
- Extensive ablation study behind the proposed idea. Considered multiple benchmarks and models.

Weakness:
- Lack of statistical error bars (CI). I think CI is important to draw conclusions from the empirical analysis, specifically in results like Table 1 where the performance of the proposed approach is close to that of the other models. LLMs are quite stochastic, and I am afraid to draw conclusions with the best case if the performances are close
- No discussion regarding the computational cost of RRMs compared to standard scalar reward models
- No discussion on limitations of the method

---

> ### Author Rebuttal · Authors · 2025-07-31
>
> We thank the reviewer for the valuable comments.
>
> ---
>
> >**Weakness 1**: Lack of statistical error bars (CI). I think CI is important to draw conclusions from the empirical analysis, specifically in results like Table 1 where the performance of the proposed approach is close to that of the other models. LLMs are quite stochastic, and I am afraid to draw conclusions with the best case if the performances are close.
>
> **Response 1**: Thank you for raising this important point. We follow the **standard evaluation protocol of RewardBench**, whose leaderboard results also do not report confidence intervals (CI).
>
> We understand the reviewer’s concern about the stochasticity of large language models and the risk of over-interpreting close performance numbers. To address this, we conducted additional experiments on RewardBench by varying random seeds. The results are shown in **Table R1**.
>
> | **Model**     | **Bench**        | **Min**   | **Max**   | **Std**    | **Reported** |
> |-----------|--------------|-------|-------|--------|----------|
> | RRM-7B    | RewardBench  | 81.9  | 83.2  | 0.005  | 82.2     |
> | RRM-32B   | RewardBench  | 90.7  | 91.4  | 0.003  | 91.2     |
>
> Table R1: Performance statistics of RRM models on RewardBench.
>
> ---
>
> >**Weakness 2**: No discussion regarding the computational cost of RRMs compared to standard scalar reward models.
>
> **Response 2**: Thank your for the thoughtful question. Unlike scalar reward models with a fixed computational budget, RRMs offer the substantial advantage of adaptively utilizing additional test-time compute. Experimental results in Figure 4 and Figure 5 empirically confirm this benefit.
>
> Moreover, the increased computational cost pertains to the training phase within RLHF, which typically involves a relatively small number of steps compared to the extensive pre-training of the language model itself. Importantly, after RLHF, RRMs do not involve the inference phase of the policy model. Therefore, the computational overhead is entirely affordable.
>
> ---
>
> >**Question 1**: What does it mean by adaptive compute? If it refers to increasing the hard budget of thinking (c.f. line 273), then it would suggest not using adaptive compute. The more or less use of the computing budget is not the result of the reasoning reward models. If there is a contribution of dynamically limiting the computation of RRM, then the adaptive compute terminology may be used.
>
> **Response 3**: RRMs adopt chain-of-thought reasoning process, explicitly signaled by the model's active decision to terminate its thought process using a \</think\> tag. Our experiments also confirm this adaptive-compute behavior by comparing the reasoning length across different domains. Specifically, we observed that the responses are shorter in the Chat domain, whereas they are considerably longer in the Reasoning domain, demonstrating dynamic allocation of computational resources based on task complexity. This reflects an emergent adaptive behavior, though it is not explicitly designed.
>
> We appreciate the reviewer’s suggestion to clarify this terminology and will revise the phrasing in the next version.
>
> ---
>
> >**Question 2**: Why didn't you consider any other models apart from Qwen?
>
> **Response 4**: Our primary focus is to address the challenge of effectively scale test-time compute for reward modeling. Given our limited GPU resources, we prioritize exploring test-time scaling behaviors and conducting experiments with large language models (e.g., 32B parameters) over repeating experiments with different base models. Since our methods are orthogonal to the base model, we plan to migrate our methods to other base models in future work. We believe the current experimental setups are sufficient to support our methods.
>
> ---
> ---
>
> Due to space limitations, we were unable to address all of **Reviewer 86j6’s** Weaknesses and Minor Weaknesses directly beneath his comments. Therefore, we provide responses to some of these points below.
>
> ---
>
> >**M9**: You report batch size and mini-batch size separately in Table 5. It is not entirely clear to me what this distinction means. Is this about gradient accumulation?
>
> **Response 18**: The distinction between batch size and mini-batch size in Table 5 follows standard GRPO training conventions and is not directly about gradient accumulation. Specifically:
>
>   - Batch size refers to the total number of sample groups collected per GRPO update round.
>
>   - Mini-batch size is the number of sample groups used in each GRPO optimization step (i.e., the batch size for backpropagation). Mini-batch size pertains to the samples per GRPO optimization step, not gradient accumulation.
>
> ---
>
> >**M10**: You should discuss that your "discriminative to scalar" scheme only works when there are sufficient candidate responses per prompt available (and mention that this generally the case in common RLHF fine-tuning scenarios).
>
> **Response 19**: Obtaining candidate responses is easy since we can always prompt LLMs to get candidates as anchors to compute Elo ratings. Besides, we would like to emphasize that such scheme enables RRMs to generate adaptive reward scores. This adaptiveness comes from their ability to utilize configurable reference candidates. Conversely, score-based reward models, by assigning absolute scores, inherently rely on an implicit and unchangeable criterion, which makes them more vulnerable to hack.

---

> > ### Comment · Reviewer_mL9a · 2025-08-01
> >
> > Thanks for your response.
> >
> > Regarding,
> > Weakness 2: No discussion regarding the computational cost of RRMs compared to standard scalar reward models.
> >
> > In the above question, instead of your defending response, it would be more helpful to get a fact-based (e.g., cubicly) or empirical response (e.g., 50 times in xyz) on how much the computation cost grows.

---

> ### Author Response · Authors · 2025-08-04
>
> Thank you very much for your positive feedback! We are happy to see our responses have addressed your concerns.
> > … instead of your defending response, it would be more helpful to get a fact-based (e.g., cubicly) or empirical response (e.g., 50 times in xyz) on how much the computation cost grows.
>
> We conducted Qwen2.5-7B post-training using feedback from RRM-32B and Skywork-Reward-Gemma-2-27B. The evaluation results on MMLU-Pro and the time used for training are as follows.
>
> | **Model**         | **Step** | **MMLU-Pro** | **Training Time** |
> |---------------|------|----------|------|
> | Qwen2.5-7B    | 0    | 47.85%   | |
> | RLHF-Skywork  | 200  | 49.51%   | |
> | RLHF-Skywork  | 400  | 48.26%   | |
> | RLHF-Skywork  | 600  | 34.76%   | 265 GPU Hours |
> | RLHF-RRM      | 200  | 50.62%   | |
> | RLHF-RRM      | 400  | 51.12%   | |
> | RLHF-RRM      | 600  | 50.86%   | 1026 GPU Hours |
>
>
> We train the models with AMD Instinct MI300 Accelerators. Training with RRM-32B and Skywork-Reward-Gemma-2-27B uses 1026 GPU hours and 265 GPU hours, which shows that computational overhead is entirely affordable using distributed training multiple GPUs.
>
> Thank you again for the valuable feedback. We will include the computation cost in the next version.

---

> > ### Comment · Reviewer_mL9a · 2025-08-04
> >
> > Thanks, I have no further questions.

---

> ### Author Response · Authors · 2025-08-05
> **Thank you for the valuable review!**
>
> Thank you so much for the valuable review. We're glad that we were able to address your concerns.

---

### Official Review · Reviewer_4QQR · 2025-06-30

**Clarity:** 3
**Significance:** 3
**Originality:** 3
**Rating:** 4
**Confidence:** 3

**Summary:**

This paper introduces Reward Reasoning Models (RRMs), which are designed to enhance reward model performance by incorporating explicit reasoning processes. The authors argue that existing reward models lack adaptability in allocating computational resources, particularly for complex queries that require nuanced analysis or multi-step reasoning. RRMs address this challenge by framing reward modeling as a reasoning task, where the model generates a chain-of-thought reasoning process before producing the final reward. This approach allows RRMs to adaptively allocate additional computational resources when evaluating complex tasks. The paper also presents a reinforcement learning framework that enables RRMs to self-evolve their reward reasoning capabilities without requiring explicit reasoning traces as training data. Experimental results demonstrate that RRMs outperform strong baselines across multiple domains, including reasoning, general knowledge, safety, and alignment with human preference. The authors further show that RRMs can effectively utilize test-time compute to improve reward accuracy and adaptively allocate computational resources to practical application scenarios.

**Questions:**

1. Can RRM effectively guide RL with unlabeled, harder mathematical and coding data, such as tasks with difficulty levels similar to AIME or LiveCodeBench?

**Ethical Concerns:**

["NO or VERY MINOR ethics concerns only"]

**Final Justification:**

While I am not very familiar with reward model literature, I maintain my current rating as the authors' responses have addressed my primary concerns. However, given that the reward model serves as a critical component, it can significantly impact RL training across multiple dimensions—including different model architectures, varying model sizes, and diverse domains.

Comprehensive evaluation would require extensive experiments to ensure that the proposed approach does not compromise the model's underlying potential. Although I recognize this as technically solid work, I believe the contribution would benefit from more thorough analysis across these various settings to fully validate the method's robustness and generalizability.

**Limitations:**

yes

**Quality:**

4

**Strengths And Weaknesses:**

**Strengths**

1. The authors conduct a comprehensive comparison between RRM and various other reward models in their experiments. The superior performance of RRM compared to other reward models highlights the effectiveness of the long reasoning reward model.

2. The work includes experiments on large reward models, such as RRM-32B, which provides more convincing conclusions about reward model training at scale.

3. The authors present a relatively thorough analysis of their proposed RRM, including BoN comparisons, RL with unlabeled data, reward model-guided DPO, parallel and sequential scaling analysis, and other detailed evaluations. These in-depth analyses further underscore the authors' significant contributions to the field of reward modeling.

---

**Weaknesses**

1. The paper lacks a deep analysis in certain model comparisons. For instance, Table 1 shows that RRM-32B outperforms Skywork-Reward-Gemma-2-27B-v0.2 on the PandaLM Test but underperforms on RewardBench. What explains these contradictory results? Similarly, in the BoN experiments in Table 2, Skywork-Reward-Gemma-2-27B-v0.2 shows significantly worse performance compared to other reward models, yet it performs strongly on RewardBench. What causes these discrepancies?

2. While the authors provide interesting further analyses, such as RL with unlabeled data, which demonstrate positive effects and training stability, they do not include the performance of other reward models in these scenarios. Would other reward models outperform RRM in tasks like RL with unlabeled data, DPO, or parallel and sequential scaling? A more detailed comparison with baselines in these areas would strengthen the work.

---

> ### Author Rebuttal · Authors · 2025-07-31
>
> We thank the reviewer for the valuable comments.
>
> ---
>
> > **Weakness 1**: The paper lacks a deep analysis in certain model comparisons. For instance, Table 1 shows that RRM-32B outperforms Skywork-Reward-Gemma-2-27B-v0.2 on the PandaLM Test but underperforms on RewardBench. What explains these contradictory results? Similarly, in the BoN experiments in Table 2, Skywork-Reward-Gemma-2-27B-v0.2 shows significantly worse performance compared to other reward models, yet it performs strongly on RewardBench. What causes these discrepancies?
>
> **Response 1**: Thank you for the insightful question. These "contradictory" results are common and expected in reward model evaluation. They arise because benchmarks vary in their data collection, domain coverage, and the specific human preferences they capture. These variations highlight the importance of evaluating reward models across diverse benchmarks. Interestingly, similar discrepancies have also been observed in concurrent work. For instance, J1[1] underperforms the Skywork reward model on RewardBench, but achieves significantly better results on the PPE benchmark.
>
> ---
>
> > **Weakness 2**: While the authors provide interesting further analyses, such as RL with unlabeled data, which demonstrate positive effects and training stability, they do not include the performance of other reward models in these scenarios. Would other reward models outperform RRM in tasks like RL with unlabeled data, DPO, or parallel and sequential scaling? A more detailed comparison with baselines in these areas would strengthen the work.
>
> **Response 2**: Thank you for recognizing our “interesting further analyses” regarding RL with unlabeled data and other aspects. We conducted additional experiments on RL post-training, where we fine-tuned Qwen2.5-7B models using feedback from both RRM-32B and Skywork-Reward-Gemma-2-27B-v0.2, respectively. The comparative results are detailed in **Table R1**. While feedback from both types of reward models improved performance on MMLU-Pro, we observed that the score-based reward model's performance quickly degraded through training, indicating a potential reward hacking issue.
>
> | **Model**         | **Step** | **MMLU-Pro** |
> |---------------|------|----------|
> | Qwen2.5-7B    | 0    | 47.85%   |
> | RLHF-Skywork  | 200  | 49.51%   |
> | RLHF-Skywork  | 400  | 48.26%   |
> | RLHF-Skywork  | 600  | 34.76%   |
> | RLHF-RRM      | 200  | 50.62%   |
> | RLHF-RRM      | 400  | **51.12%**   |
> | RLHF-RRM      | 600  | 50.86%   |
>
> Table R1: Post-training MMLU-Pro results for Qwen2.5-7B using feedback from RRM-32B vs. Skywork-Reward-Gemma-2-27B. RRM yields stable gains, while the score-based model shows degradation, indicating potential reward hacking.
>
> ---
>
> > **Question 1**: Can RRM effectively guide RL with unlabeled, harder mathematical and coding data, such as tasks with difficulty levels similar to AIME or LiveCodeBench?
>
> **Response 3**: Yes. We have conducted test-time reinforcement learning (TTRL) experiments, where we train DeepScaleR-1.5B-Preview[2] on unlabeled queries of AIME24. As shown in **Table R2**, RRMs successfully guided the model to improve its performance on difficult math questions.
>
> |**Model**                                    | **AIME24** |
> |------------------------------------------|--------|
> | DeepScaleR-1.5B-Preview                  | 43.3   |
> | DeepScaleR-1.5B-Preview + RRM-32B TTRL   | **50.0**   |
>
> Table R2: Pass@1 on AIME24 using 5 samples for each question. RRM-guided training improves DeepScaleR-1.5B-Preview performance on challenging math problems without using labeled data.
>
> ---
>
> **References**
>
> **[1]** J1: Incentivizing Thinking in LLM-as-a-Judge via Reinforcement Learning
>
> **[2]** DeepScaleR: Surpassing O1-Preview with a 1.5B Model by Scaling RL
>
> ---
> ---
>
> Due to space limitations, we were unable to address all of **Reviewer 86j6’s** Weaknesses and Minor Weaknesses directly beneath his comments. Therefore, we provide responses to some of these points below.
>
> ---
>
> >**M1**: I would expect that the batch-ELO based scalar reward derivation has high variance, depending on the batch. It's nice to see that it works nonetheless. It would be even better to have a dedicated evaluation of this, for example by using a natively scalar reward model from which you can derive a discriminative one, which you can then turn back into a scalar one with your proposed approach -- what is the loss here? Another way to approach this would be to compare DPO with ELO+RLHF. I understand that this is not your primary contribution, however, so this does not impact my score much.
>
> **Response 10**: We appreciate the reviewer’s suggestions. In our setup, the ELO-based scalar scores are derived by ranking the model’s own generated responses, which are then used to train a policy via RL. To address the reviewer’s concern about variance, we conducted a simple empirical test: in a setting with 8 candidate responses, we let each candidate play against 4 randomly selected opponents, and repeated this process 1,000 times. We then computed the standard deviation of the resulting ELO scores across trials. The relative standard deviation (std/mean) for all responses was below 1%, indicating that the batch-based ELO estimates are in fact quite stable. The results are summarized in the following **Table R3**.
>
> | **Candidate** | **Relative Standard Deviation** |
> |-----------|-----------------------------|
> | 1         | 0.197%                      |
> | 2         | 0.259%                      |
> | 3         | 0.244%                      |
> | 4         | 0.288%                      |
> | 5         | 0.195%                      |
> | 6         | 0.002%                      |
> | 7         | 0.247%                      |
> | 8         | 0.285%                      |
>
> Table R3: The relative standard deviation of each candidates.
>
> ---
>
> >**M2**: Why do you use voting@16 in Table 1, voting@5 in Table 2?
>
> **Response 11**: We've verified that our core conclusions remain consistent whether using voting@5 or voting@16. In cases with 32 candidates for each question in Table 2, we use voting@5 to manage computational load effectively while still demonstrating the benefits of parallel scaling, as our primary aim was to show performance improvement rather than tune voting size for each specific setup.
>
> ---
>
> >**M3**: How is the CI computed in Table 3? I am surprised that it is not symmetrical.
>
> **Response 12**: The confidence intervals in Table 3 are computed by following the Arena-Hard protocol, using non-parametric bootstrap resampling, which does not assume symmetry in the underlying accuracy distribution. Thus, the CI bounds can be asymmetric. For reference, the official Arena leaderboard also reports asymmetric confidence intervals.
>
> ---
>
> >**M4**: The paper (especially the RW) seems to imply that generative RMs are always discriminative. I would have thought those are two distinct axes (generative RMs use the full decoder and a textual response, but may still be either scalar or discriminative). I am not perfectly familiar with established terminology, could you elaborate? LLM-Blender might be a good reference to discuss discriminative RMs, but that is only one possible example and you are of course not required to cite it.
>
> **Response 13**:  As mentioned in Related Work, **reward formulation (scalar vs. generative) and scoring scheme (individual vs. discriminative)** are two distinct axes. We clarify this distinction in lines 312–314: ‘Reward models can be characterized along two dimensions: reward formulation and scoring scheme’ and in lines 327–328: ‘with flexibility for both single-instance evaluation and multi-response comparison.’ So **generative reward models are not necessarily discriminative** — they can operate in an individual fashion by taking a single input (e.g., a prompt–response pair) and producing a justification.
> Besides, note that we have already cited **LLM-Blender** in the paper, and we agree that it is indeed a relevant example.
>
> ---
>
> >**M5**: The RW section does not explicitly put RRM into context of the related works ("compare and contrast"). This is largely implied, so this is a very minor point, but it would be better to make it explicit.
>
> **Response 14**: Thanks for your suggestion. We will revise the section to more clearly highlight the distinctions and connections between our approach and existing reward models.
>
> ---
>
> >**M6**: How does such a discriminative RM contrast to LLM-as-a-judge? Is there any difference?
>
> **Response 15**: “LLM-as-a-judge” usually refers to using a general-purpose LLM to judge responses without explicit training for reward modeling. It typically works in a zero-shot or few-shot manner via prompting. Differently, discriminative RMs focus on reward modeling, and are often more accurate and stable on specific domains.
>
> ---
>
> >**M7**: The paper skips over some preliminaries: What is post-training? On line 31: Decoding layer of what?
>
> **Response 16**: Post-training refers to a training stage after pre-training. In our paper, post-training can refer to either training RRMs or training policy models with RRM feedback.
>
> "Decoding layer" refers to the language modeling head (i.e., the final projection layer) of LM. Scalar reward models typically replace the head with a linear layer that outputs a scalar reward instead of token logits. We will revise the wording to make this clearer.
>
> ---
>
> >**M8**: What do you mean by the "round-robin tournament structure"?
>
> **Response 17**: By "round-robin tournament structure", we mean that all response candidates are compared against each other in all possible pairs. For $N$ candidates, this results in $\binom{N}{2}$ pairwise comparisons.

---

> > ### Comment · Reviewer_4QQR · 2025-08-05
> >
> > Thank you for the detailed supplement. These results address most of my concerns.

---

> ### Author Response · Authors · 2025-08-07
> **Thank you for the valuable review!**
>
> Thank you so much for the valuable review. We're glad that we were able to address your concerns.

---

### Official Review · Reviewer_86j6 · 2025-07-02

**Clarity:** 3
**Significance:** 4
**Originality:** 3
**Rating:** 5
**Confidence:** 4

**Summary:**

The paper introduces reward reasoning models, an approach to train generative preference models with reasoning capabilities using reinforcement learning. They fine-tune a base reasoning language model using reinforcement learning on a preference dataset with a rule-based reward rewarding correct preference prediction. Reasoning emerges (or further develops from the reasoning capabilities of the base model) as a way to leverage test-time compute and improve preference prediction quality, resulting in larger reward for the model.

As a secondary contribution, the paper introduces an approach based on in-batch ELO to use pairwise preference models in RL fine-tuning pipelines that expect scalar rewards.

The authors train multiple reward models (7B, 14B, 32B) using this approach and evaluate them thoroughly, both on direct reward modeling benchmarks and downstream tasks.

**Questions:**

Please see the numbered weaknesses, focusing on the major ones (**WX**). Most of them are actionable, i.e., can be considered questions, although many of the minor ones (**MX**) do not require an individual response. Actual questions regarding the understanding of the paper are listed as weaknesses in the clarity section. The minor points have little impact on my score.

The paper is high-quality and significant. I would consider raising my score if the clarity is somewhat improved. Of particular importance are: W2, W3, and W4.

**Ethical Concerns:**

["NO or VERY MINOR ethics concerns only"]

**Final Justification:**

The suggested reasoning approach is very promising for further improving language model validation and training. The writing and the evaluation are good. The authors have addressed all the points I raised.

**Limitations:**

- **W9** The experiments show that there are many types of tasks where reasoning does *not* significantly improve performance (e.g., Table 1 Chat). It still likely has some overhead, even on those tasks. This should be discussed, and ideally the overhead analyzed (is the compute allocation really adaptive, i.e., does it use less on those tasks?)
- **M10** You should discuss that your "discriminative to scalar" scheme only works when there are sufficient candidate responses per prompt available (and mention that this generally the case in common RLHF fine-tuning scenarios).

**Paper Formatting Concerns:**

The concerns below are listed as a service to you; they do not impact my score.

- Figure 1: scaler -> scalar
- L96: relies -> rely
- L255: perform -> performance
- L256: make -> making
- L271: Unintended line break.

**Quality:**

4

**Strengths And Weaknesses:**

In the following, I discuss strength and weaknesses on the dimensions of quality, clarity, significance, and originality.
Weaknesses are marked with a **W**, minor weaknesses with an **M**, notes that do not impact my score with **N**, and strengths with an **S**.
Weaknesses are numbered for reference purposes.

## Quality

- **S** The RRM models are evaluated extensively on a variety of benchmarks.
- **W1** It would be good to discuss related work on general reasoning and parallel test-time compute strategies (e.g., majority vote, knockout tournament).
- **W2** The evaluation in Table 2 seems slightly unfair to me, as all the baseline reward models are pure preference models (correct?) focused mostly on style, while your model has verifiable tasks in the training data. It would be very informative to have DirectJudge in this table.
- **M1** I would expect that the batch-ELO based scalar reward derivation has high variance, depending on the batch. It's nice to see that it works nonetheless. It would be even better to have a dedicated evaluation of this, for example by using a natively scalar reward model from which you can derive a discriminative one, which you can then turn back into a scalar one with your proposed approach -- what is the loss here? Another way to approach this would be to compare DPO with ELO+RLHF. I understand that this is not your primary contribution, however, so this does not impact my score much.
- **M2** Why do you use voting@16 in Table 1, voting@5 in Table 2?
- **N** The description of Table 6 in the appendix is slightly confusing. It sounds like it should have numerical results matching Figure 10, but the results differ (RRM-7B ELO instead of Knockout tournament).
- **N** A brief discussion of what PRM is would be useful in Appx B.1.2.
- **N** You always remove reasoning traces from your training data. Doesn't that mean that reasoning is then out-of-distribution for the trained RM? I would expect that it may further improve performance if you leave some reasoning traces in.
- **N** It would be interesting to see if you can use the budget forcing approach introduced by S1 (https://arxiv.org/abs/2501.19393) to further improve performance. (Just a note, does not impact my score.)

## Clarity

- **S** The paper is well-written and easy to follow.
- **W3** Page 8 mentions a 14B model, but you previously only discuss how you trained the 7B and 32B ones. Table 5 also only has hyperparameters for those.
- **W4** Your RRMs were fine-tuned solely with RL, no SFT right? How many RL steps?
- **W5** The "reward model" terminology may be confusing, as your model does not directly produce rewards but rather choices. It would be more precise to call it a preference model, choice model, or binary classifier. Discriminative reward model (as you use in the paper) could also work as a "compromise". Is there precedent for calling a choice model a reward model? If so, I understand if you want to stick to this terminology but you should make this clearer by (1) discussing early on that you propose a discriminative model and (2) take particular care to avoid confusion everywhere you mention "reward" (such as in line 47, Figure 1, line 216). I think this may easily confuse a reader.
- **W6** I do not understand what is evaluated on Figure 4. Number of pairs used for what?
- **W7** I do not understand what is measured in Table 4. What is accuracy in the context of Tournament / ELO rating?
- **W8** I would appreciate an analysis of the overhead generated by the ELO-based reward derivation (if possible without extensive new experiments).
- **M3** How is the CI computed in Table 3? I am surprised that it is not symmetrical.
- **M4** The paper (especially the RW) seems to imply that generative RMs are always discriminative. I would have thought those are two distinct axes (generative RMs use the full decoder and a textual response, but may still be either scalar or discriminative). I am not perfectly familiar with established terminology, could you elaborate? LLM-Blender (https://arxiv.org/abs/2306.02561) might be a good reference to discuss discriminative RMs, but that is only one possible example and you are of course not required to cite it.
- **M5** The RW section does not explicitly put RRM into context of the related works ("compare and contrast"). This is largely implied, so this is a very minor point, but it would be better to make it explicit.
- **M6** How does such a discriminative RM contrast to LLM-as-a-judge? Is there any difference?
- **M7** The paper skips over some preliminaries: What is post-training? On line 31: Decoding layer of what?
- **M8** What do you mean by the "round-robin tournament structure"?
- **M9** You report batch size and mini-batch size separately in Table 5. It is not entirely clear to me what this distinction means. Is this about gradient accumulation?
- **N** The colors in Figure 6 (light orange / pink / dark orange / brown) are hard to distinguish when printed.
- **N** I think it would be worth pointing out that the ELO-based approach is generally applicable to any discriminative model.

## Significance

With increasing attention being paid to scaling up inference compute for LLMs, scaling up verification is of increasing significance. This paper is a step into this direction.


## Originality

Exempting concurrent work, the paper is novel to the best of my knowledge. I am not perfectly familiar with the related work, however. At its core it is a fairly straightforward application of the general "reasoning recipe" to reward modeling, tempering originality slightly, but it is well-executed and evaluated and therefore a valuable contribution to the literature.

---

> ### Author Rebuttal · Authors · 2025-07-31
>
> Thank you for your feedback, particularly for recognizing the "high quality and significance" of our paper and for considering raising the score. We deeply appreciate your valuable comments regarding clarity, and we are committed to refining the paper.
>
> Due to space constraints, we prioritize addressing the main **Weaknesses** below. For points categorized as **Minor Weaknesses**, we will respond M1-M8 under **the reviewer 4QQR** and M9-M10 under **the reviewer mL9a**.
>
> Regarding the **Notes**, we appreciate your insightful notes, which will improve the quality and clarity of our paper. Due to length constraints, we cannot provide individual responses, but we will revise the paper accordingly.
>
> ---
>
> >**W1**: It would be good to discuss related work on general reasoning and parallel test-time compute strategies (e.g., majority vote, knockout tournament).
>
> **Response 1**: Thank you for the suggestion. We cited parallel test-time compute strategies in Related Work (Line 339-341). We will expand the discussion in related work in the next version.
>
> ---
>
> >**W2**: The evaluation in Table 2 seems slightly unfair to me, as all the baseline reward models are pure preference models (correct?) focused mostly on style, while your model has verifiable tasks in the training data. It would be very informative to have DirectJudge in this table.
>
> **Response 2**: The Preference Proxy Evaluations (PPE) benchmark used in Table 2 is a standard protocol for evaluating reward models,  and is a widely accepted evaluation setup in the reward modeling literature.
>
> Furthermore, the baseline reward models in Table 2 are not purely style-focused. They are trained on preference data that also includes **verifiable reasoning tasks**. For example, the training datasets of **Skywork-Reward, DeepSeek-GRM, and J1**[1] all  contains verifiable reasoning tasks.
>
> We understand the reviewer’s concern about potential differences in training data and evaluation setup. To address this, we additionally follow the protocol from the PPE paper[2] and evaluate reward models on **binary preference classification**, as shown in Table R1. This evaluation includes a broader set of baselines under the same setup. Due to response length limitation, please refer to Table 2 of the PPE’s paper[2] for more baselines.
>
> | **Model**                             | **MMLU-Pro** | **MATH**  | **GPQA**  | **Overall** |
> |-----------------------------------|----------|--------|--------|---------|
> | Skywork-Reward-Gemma-2-27B        | 55.0     | 46.2   | 44.7   | 48.6    |
> | Gemma-2-27B                       | 66.2     | 66.4   | 51.9   | 61.5    |
> | DeepSeek-GRM-27B (voting@32)      | 65.5     | 69.4   | 56.0   | 63.6    |
> | DeepSeek-GRM-27B (MetaRM)         | 68.1     | 70.0   | 56.9   | 65.0    |
> | Llama-3.1-8B-Instruct             | 56.3     | 62.9   | 51.4   | 56.9    |
> | Llama-3.1-70B-Instruct            | 72.1     | 73.1   | 61.2   | 68.8    |
> | J1-Llama-8B (SC@32)               | 67.5     | 76.6   | 55.7   | 66.7    |
> | J1-Llama-70B (SC@32)              | 79.9     | 88.1   | 66.5   | 78.2    |
> | **RRM-7B**                            | 66.5     | 88.0   | 57.9   | 70.3    |
> | **RRM-7B (voting@5)**                 | 68.3     | 90.5   | 58.3   | 72.4    |
> | **RRM-14B**                           | 71.8     | 88.6   | 60.2   | 73.5    |
> | **RRM-14B (voting@5)**                | 72.4     | 89.7   | 61.1   | 74.4    |
> | **RRM-32B**                           | 80.5     | 94.3   | 67.4   | 80.7    |
> | **RRM-32B (voting@5)**                | **81.3**     | **95.4**   | **68.4**   | **81.7**    |
>
> Table R1: Evaluation results on binary preference classification following the protocol from Frick et al [2]. We report accuracy over five pairs of conflicted responses.
>
> ---
>
> >**W3**: Page 8 mentions a 14B model, but you previously only discuss how you trained the 7B and 32B ones. Table 5 also only has hyperparameters for those.
>
> **Response 3**: Initially, we focused on training **7B and 32B** reward models to align with the model sizes of prior work. However, to better illustrate the **scaling trends** in reward model performance, we later added a 14B model as an intermediate point. Table R.1 provides evaluation results of RRM-14B.
>
> We appreciate your careful review and will provide the hyperparameters and additional evaluation results for the 14B model in the appendix.
>
> ---
>
> >**W4**: Your RRMs were fine-tuned solely with RL, no SFT right? How many RL steps?
>
> **Response 4**: Yes. RRMs were fine-tuned solely with RL, with the training steps being 720 and 440 for the 7B model and the 32B model respectively.
>
> ---
>
> >**W5**: The "reward model" terminology may be confusing, as your model does not directly produce rewards but rather choices. It would be more precise to call it a preference model, choice model, or binary classifier. Discriminative reward model (as you use in the paper) could also work as a "compromise". Is there precedent for calling a choice model a reward model? If so, I understand if you want to stick to this terminology but you should make this clearer by (1) discussing early on that you propose a discriminative model and (2) take particular care to avoid confusion everywhere you mention "reward" (such as in line 47, Figure 1, line 216). I think this may easily confuse a reader.
>
> **Response 5**: While our model produces pairwise choices rather than scalar rewards, the terminology of “reward model” for such discriminative models has been adopted in prior work. For example, GenRM[3]  uses this term and illustrates it in Figure 8 in prompt structure. Similarly, PairJudge RM[4] presents a similar setup in their prompt table (Table 4), yet still refers to the model as a reward model.
>
> We agree this terminology may be confusing for readers unfamiliar with this convention. We will revise the terminology accordingly to enhance clarity.
>
> ---
>
> >**W6**: I do not understand what is evaluated on Figure 4. Number of pairs used for what?
>
> **Response 6**: For each MATH problem, we sample 8 candidate responses and select several pairs from them for pairwise comparison. These comparison results are then used for ELO rating to estimate a ranking over the 8 candidates. The top-ranked response is selected as the final output. The y-axis in Figure 4 represents the **accuracy** of this selected output. The number of pairs reflects the test-time computational budget for comparisons. Comparing all $\binom{8}{2} = 28$ possible pairs represents a full round-robin comparison. Additionally, the rightmost point in Figure 4 signifies 140 pairwise comparisons (28 pairs each compared 5 times by voting).
>
> ---
>
> >**W7**: I do not understand what is measured in Table 4. What is accuracy in the context of Tournament / ELO rating?
>
> **Response 7**: Both Tournament and ELO rating can be used to select the best candidate among a set of responses, based on the RRM's pairwise comparisons. The accuracy reported in Table 4 refers to the **correctness of the top-selected response** under each strategy.
>
> ---
>
> >**W8**: I would appreciate an analysis of the overhead generated by the ELO-based reward derivation (if possible without extensive new experiments).
>
> **Response 8**: The ELO-based reward derivation introduces additional computational overhead primarily due to the pairwise comparisons required to estimate rankings among candidate responses. For a batch size of $N\$ candidates, the number of pairwise matches is $O(N \times k)\$, where $k\$ is a hyperparameter representing the number of competitors each candidate is compared against (e.g., 4 in our setup). In practice, we observed that this overhead remains manageable and does not dominate the overall training time, as the pairwise scoring is efficiently parallelized.
>
> ---
>
> >**W9**: The experiments show that there are many types of tasks where reasoning does not significantly improve performance (e.g., Table 1 Chat). It still likely has some overhead, even on those tasks. This should be discussed, and ideally the overhead analyzed (is the compute allocation really adaptive, i.e., does it use less on those tasks?)
>
> **Response 9**: Thank you for the valuable suggestion. To analyze the potential overhead, we computed the average thinking length across different task subtypes in RewardBench, as **Table R2** shows. We observe that tasks like Chat have relatively short responses, indicating less computational overhead, while reasoning tasks produce much longer responses. This suggests that the compute allocation is indeed adaptive.
>
> | **Task Subtype** | **Average Thinking Length (Tokens)** |
> |--------------|----------------------------------|
> | Chat         | 421.97                           |
> | Chat Hard    | 363.71                           |
> | Safety       | 319.62                           |
> | Reasoning    | 848.13                           |
>
> Table R2: Average number of tokens before `</think>` across task subtypes.
>
> ---
>
> **References**
>
> **[1]** J1: Incentivizing Thinking in LLM-as-a-Judge via Reinforcement Learning
>
> **[2]** How to evaluate reward models for RLHF
>
> **[3]** Generative Reward Models
>
> **[4]** PairJudge RM: Perform Best-of-N Sampling with Knockout Tournament

---

> > ### Comment · Reviewer_86j6 · 2025-08-04
> >
> > Thank you for your detailed response, adequately addressing all the points I raised. As most comments were on the clarity side, I think the paper would be improved (beyond it's already good state) if you incorporate these clarifications in the final version.
> >
> > Given no outstanding issues, I will raise my score to recommend acceptance (5).

---

> ### Author Response · Authors · 2025-08-04
> **Thank you for raising the score!**
>
> Thank you again for providing detailed constructive comments and for raising the score! We will integrate these clarifications and the additional experiment results in the subsequent version.

---

### Official Review · Reviewer_DJnD · 2025-07-02

**Clarity:** 2
**Significance:** 2
**Originality:** 3
**Rating:** 3
**Confidence:** 3

**Summary:**

This paper introduced reward reasoning model (RRM), reward model that can generated reasoning about the given generation. They train the reasoning policy via reinforcement fine-tuning (i.e. GRPO) and further boost performance through parallel test-time scaling (e.g., voring over multiple comparisons). The paper evaluates RRMs on multiple ways including, reward benchmark, best-of-N selection, preference alignment, and RL fine-tuning, claiming consistent gains and practical utility.

**Questions:**

[Q1] Can you show that RRM-based RL fine-tuning improves performance for other base models or datasets beyond DeepSeek-R1-Distill-Qwen-7B and WebInstruct?

[Q2] When using RRM for RL, please compare RRM with other baselines such as Direct Judge or other outcome reward models.

[Q3] Please suggest more explanation of what is the RRMs’ advantage against other concurrent methods.

**Ethical Concerns:**

["NO or VERY MINOR ethics concerns only"]

**Final Justification:**

Some of my concerns have been resolved after the rebuttal, including points regarding the inconsistent improvements on the main RewardBench, clarification on post-training evaluation, and differences with concurrent works.

However, I still find the post-training evaluation insufficient, especially regarding how the reward model is actually used for RLHF and the trade-off between training time and performance. (authors added more experiments, but this is hard to resolve in insufficient time). The authors responded that RLHF is not the only intended usage, which may be controversial. \
Additionally, regarding the issue of training time versus performance trade-off, the authors claim that this is not a key focus of their research; however, I believe that further discussion would have been beneficial.

While some concerns have been addressed, others remain unresolved. Thus, I am increasing my score to a borderline reject.

**Limitations:**

The paper does not discuss originality and technical novelty against other concurrent works and reasoning-trained models.

**Paper Formatting Concerns:**

No issues were found.

**Quality:**

2

**Strengths And Weaknesses:**

[S1] Effective CoT-based RL fine-tuning: The paper demonstrates that training long chain of thought (CoT)  reasoning through reinforcement learning can help reward models to enhance the performance of reward model.

[W1] Inconsistent improvements: In the main RewardBench table, RRM without voting does not clearly outperform strong baselines such as DirectJudge; observed gains are small and sometimes statistically insignificant.

[W2] Narrow post-training evaluation: The RL experiments use only a single setting (e.g. DeepSeek-R1-Distill-Qwen-7B on WebInstruct with GRPO). Also, the study lacks comparisons against other baselines (i.e. DirectJudge, score-based RM). It makes it hard to judge whether RRM feedback is uniquely beneficial.

[W3] Lack of technical novelty: The suggested concurrent work combines GRPO with additional technical novelty and often reports stronger results against RRM. RRM relies mainly on standard RL training suggested by Deepseek, which may limit its novelty.

---

> ### Author Rebuttal · Authors · 2025-07-31
>
> Thank you for your comments.
>
> ---
>
> >**Weakness 1**: Inconsistent improvements: In the main RewardBench table, RRM without voting does not clearly outperform strong baselines such as DirectJudge; observed gains are small and sometimes statistically insignificant.
>
> **Response 1**: We appreciate the reviewer’s close analysis of the results in the main RewardBench table. Our primary contribution is not to incrementally improve upon a single benchmark, but rather to propose a new paradigm for reward modeling — effectively scales with test-time compute in reward modeling. The reviewer correctly notes that RRM without voting does not always significantly outperform DirectJudge. This observation, in fact, highlights a key strength of our framework. RRM can leverage both **sequential scaling** and **parallel scaling** to improve the performance.
>
> Most importantly, we respectfully disagree with the characterization of our improvements as "inconsistent." When viewed across the full suite of evaluations, RRMs demonstrate remarkably consistent gains. As shown in our paper, RRMs achieve competitive performance on a wide range of benchmarks **beyond RewardBench, including PandaLM Test, MMLU-Pro, MATH, GPQA, and Arena-Hard**.
>
> ---
>
> >**Weakness 2**: Narrow post-training evaluation: The RL experiments use only a single setting (e.g. DeepSeek-R1-Distill-Qwen-7B on WebInstruct with GRPO). Also, the study lacks comparisons against other baselines (i.e. DirectJudge, score-based RM). It makes it hard to judge whether RRM feedback is uniquely beneficial.
>
> **Response 2**: Our primary focus is to address the challenge of effectively scale test-time compute for reward modeling. To this end, we conducted extensive evaluations including agreement with human preferences, performance in best-of-N inference across multiple domains, and detailed scaling law analyses, which form our contribution in terms of experiments.
>
> In response to the reviewer's suggestion, we have conducted **additional RL post-training experiments**.
>
>   - We now include experiments fine-tuning a different, publicly available base model, **Qwen2.5-7B**.
>
>   - We compare the performance of fine-tuning with our **RRM-32B** against a strong, score-based reward model baseline, **Skywork-Reward-Gemma-2-27B-v0.2**. The new results, presented in **Table R1**.
>
> | **Model**         | **Step** | **MMLU-Pro** |
> |---------------|------|----------|
> | Qwen2.5-7B    | 0    | 47.85%   |
> | RLHF-Skywork  | 200  | 49.51%   |
> | RLHF-Skywork  | 400  | 48.26%   |
> | RLHF-Skywork  | 600  | 34.76%   |
> | RLHF-RRM      | 200  | 50.62%   |
> | RLHF-RRM      | 400  | **51.12%**   |
> | RLHF-RRM      | 600  | 50.86%   |
>
> Table R1: Post-training MMLU-Pro results for Qwen2.5-7B using feedback from RRM-32B vs. Skywork-Reward-Gemma-2-27B. RRM yields stable gains, while the score-based model shows degradation, indicating potential reward hacking.
>
> We also additionally conducted test-time reinforcement learning (TTRL) experiments, where we train **DeepScaleR-1.5B-Preview**[1] on unlabeled queries of AIME24. As shown in **Table R2**, RRMs successfully guided the model to improve its performance on difficult math questions.
>
> |**Model**                                    | **AIME24** |
> |------------------------------------------|--------|
> | DeepScaleR-1.5B-Preview                  | 43.3   |
> | DeepScaleR-1.5B-Preview + RRM-32B TTRL   | **50.0**   |
>
> Table R2: Pass@1 on AIME24 using 5 samples for each question. RRM-guided training improves DeepScaleR-1.5B-Preview performance on challenging math problems without using labeled data.
>
> ---
>
> >**Weakness 3**: Lack of technical novelty: The suggested concurrent work combines GRPO with additional technical novelty and often reports stronger results against RRM. RRM relies mainly on standard RL training suggested by Deepseek, which may limit its novelty.
>
> **Response 3**: We would appreciate it if the reviewer could specify the "suggested concurrent work", as we were unable to identify any concurrent work in your comment.
>
> Regarding the claim that concurrent work "often reports stronger results", the results in **Table R3 and R4** show that RRM consistently outperform concurrent models , consisting of DeepSeek-GRM, RM-R1, and J1[2], on both RewardBench and the (PPE) benchmark.
>
> | **Model**                                           | **Chat**  | **Chat Hard** | **Safety** | **Reasoning** | **Overall** |
> |------------------------------------------------|-------|-----------|--------|-----------|---------|
> | DeepSeek-GRM-27B                         | 94.1  | 78.3      | 88.0   | 83.8      | 86.0    |
> | RM-R1-Distill-Qwen-7B                         | 88.9  | 66.2      | 78.4   | 87.0      | 80.1    |
> | RM-R1-Distill-Qwen-32B                         | 95.3  | 80.3      | 91.1   | 96.8      | 90.9    |
> | **RRM-7B**                      | 87.7  | 70.4      | 80.7   | 90.0      | 82.2    | 72.9           | 71.1           |
> | **RRM-7B (voting@16)**          | 92.1  | 71.5      | 81.3   | 93.8      | 84.8    | 75.9           | 77.8           |
> | **RRM-32B**                     | 94.7  | 81.1      | 90.7   | 98.3      | 91.2    | 78.8           | 79.0           |
> | **RRM-32B (voting@16)**        | **96.1**  | **81.4**      | **91.6**   | **98.6**      | **91.9**
>
> Table R3: Performance of RRM compared to concurrent methods on RewardBench.
>
> | **Model**                             | **MMLU-Pro** | **MATH**  | **GPQA**  | **Overall** |
> |-----------------------------------|----------|--------|--------|---------|
> | DeepSeek-GRM-27B (voting@32)      | 65.5     | 69.4   | 56.0   | 63.6    |
> | DeepSeek-GRM-27B (MetaRM)         | 68.1     | 70.0   | 56.9   | 65.0    |
> | J1-Llama-8B (SC@32)               | 67.5     | 76.6   | 55.7   | 66.7    |
> | J1-Llama-70B (SC@32)              | 79.9     | 88.1   | 66.5   | 78.2    |
> | **RRM-7B**                            | 66.5     | 88.0   | 57.9   | 70.3    |
> | **RRM-7B (voting@5)**                 | 68.3     | 90.5   | 58.3   | 72.4    |
> | **RRM-14B**                           | 71.8     | 88.6   | 60.2   | 73.5    |
> | **RRM-14B (voting@5)**                | 72.4     | 89.7   | 61.1   | 74.4    |
> | **RRM-32B**                           | 80.5     | 94.3   | 67.4   | 80.7    |
> | **RRM-32B (voting@5)**                | **81.3**     | **95.4**   | **68.4**   | **81.7**   |
>
> Table R4: Performance of RRM compared to concurrent methods on PPE.
>
> We respectfully disagree with the assertion that our method's novelty is limited because it "relies mainly on standard RL training." Our core technical contribution is not the invention of a new RL algorithm (like GRPO) or a new architecture (like the Transformer). Rather, our novelty lies in **the insight and formulation of performing explicit reasoning to adaptively allocate compute for reward modeling**. This new paradigm is what allows RRMs to scale effectively and achieve superior performance on a wide range of benchmarks. Foundational methods are the building blocks of scientific progress; leveraging them to create a novel and effective system should not be a weakness.
>
> ---
>
> >**Question 1**:  Can you show that RRM-based RL fine-tuning improves performance for other base models or datasets beyond DeepSeek-R1-Distill-Qwen-7B and WebInstruct?
>
> **Response 4**: We conduct additional evaluations on RL post-training, where we post-train Qwen2.5-7B models with RRM-32B and Skywork-Reward-Gemma-2-27B-v0.2, respectively. We also post-train DeepScaleR-1.5B-Preview with RRM-32B **on the math domain**. Please find the results in **Table R1 and R2**.
>
> ---
>
> >**Question 2**: When using RRM for RL, please compare RRM with other baselines such as DirectJudge or other outcome reward models.
>
> **Response 5**: We conduct additional evaluations on RL post-training, where we post-train Qwen2.5-7B models with RRM-32B and Skywork-Reward-Gemma-2-27B-v0.2, respectively. We also post-train DeepScaleR-1.5B-Preview with RRM-32B **on the math domain**. Please find the results in **Table R1 and R2**. It's important to clarify that DirectJudge is not a typical score-based RM but an ablation of RRM as baseline, thus existing methods cannot directly incorporate DirectJudge into RL post-training.
>
> ---
>
> >**Question 3**: Please suggest more explanation of what is the RRMs’ advantage against other concurrent methods.
>
> **Response 6**:
>   - **Simplicity and Efficiency**: RRM provides a unified and straightforward framework. In contrast, methods like DeepSeek-GRM introduce a complicated, multi-stage workflow requiring several intermediate models before producing the final reward model. Our end-to-end approach is significantly simpler to implement and train.
>
>   - **Performance and Scaling**: As shown in **Table R3 and R4**, our RRM outperforms DeepSeek-GRM-27B on both RewardBench and PPE. More critically, RRM demonstrates better scaling properties. It shows that DeepSeek-GRM-27B struggles to benefit from increased test-time compute on difficult reasoning benchmarks like MMLU-Pro and GPQA, whereas RRM achieves consistent and significant gains from parallel scaling on these same tasks. This demonstrates RRM's superior ability to tackle complex evaluation queries.
>
> ---
>
> **Reference**
>
> **[1]** DeepScaleR: Surpassing O1-Preview with a 1.5B Model by Scaling RL
>
> **[2]** J1: Incentivizing Thinking in LLM-as-a-Judge via Reinforcement Learning

---

> > ### Comment · Reviewer_DJnD · 2025-08-04
> >
> > Thank you for your rebuttal. While some of my concerns have been addressed by your responses, I have a few follow-up questions.
> >
> > **[W1]**
> > Thank you for clarifying that RRM outperforms baselines in best-of-N inference evaluations. I have a follow-up question which I hope can improve the robustness of the overall evaluation:
> >
> > **[W1-1]** Why does the consistency across benchmarks not hold? For example, in Table 1, Skywork-Reward-Gemma-2-27B-v0.2 is the strongest model on RewardBench, yet it lags far behind RRM in RLHF fine-tuning. Could you provide some insight into this discrepancy?
> >
> > **[W2]**
> > Thank you for conducting additional experiments. Since the most critical application of reward models (RMs) is in RLHF, I believe comparing RRM with baselines in this context is essential. I have some further questions about your evaluation:
> >
> > **[W2-1]** Although RRM outperforms Skywork on Qwen2.5-7B by 1.66% and 4.27%, this improvement seems smaller compared to the results shown with PPE (noting that the original PPE experiment in the draft used GPT-4o). Would it be possible to provide PPE results with Qwen as well, to clarify the correlation?
> >
> > **[W2-2]** In your RLHF experiments, Skywork’s performance increases at 200 steps but drops at 400 and 600 steps, suggesting that there may be a sweet spot between 0 and 400 steps. By the same logic, there may be an optimal checkpoint for RLHF-RRM between 200 and 600 steps. Providing results for intermediate steps could further enhance the robustness of your evaluation.
> >
> > **[W2-3]** I am also interested in the trade-off between compute budget and performance. Scalar RMs and general Generative RMs are efficient but less performant, while RRM can adaptively improve performance by increasing compute via parallel sampling or longer reasoning. However, inference efficiency is important in practice, since the compute budget for post-training is often limited. Could you provide a comparison of performance versus compute budget (e.g., inference FLOPs + training FLOPs or elapsed time during RL), including other baselines and varying the test-time scaling of RRM?
> >
> > **[W3, Q1, Q2, Q3]**
> > Thank you for addressing these points.

---

> ### Author Response · Authors · 2025-08-04
>
> We are glad to hear that our responses have addressed your concerns regarding **technical novelty** and **comparison with concurrent work**. We will now address the additional weaknesses raised in your latest comments.
>
> > Why does the consistency across benchmarks not hold? For example, in Table 1, Skywork-Reward-Gemma-2-27B-v0.2 is the strongest model on RewardBench, yet it lags far behind RRM in RLHF fine-tuning. Could you provide some insight into this discrepancy?
>
> Our results highlight a critical observation: performance on a static benchmark like RewardBench does not always correlate with a reward model's effectiveness in RLHF fine-tuning. This is precisely the reason why we conducted the RL post-training/DPO experiments, which are often **omitted in concurrent work** such as Deepseek-GRM, J1, and RM-R1. Therefore, demonstrating the effectiveness of our approach through RL post-training is a **contribution, not a weakness**.
>
> > Although RRM outperforms Skywork on Qwen2.5-7B by 1.66% and 4.27%, this improvement seems smaller compared to the results shown with PPE (noting that the original PPE experiment in the draft used GPT-4o). Would it be possible to provide PPE results with Qwen as well, to clarify the correlation?
>
> The results from PPE and MMLU-Pro are **not comparable** because they represent entirely different evaluation settings. As detailed in the PPE paper [1], the PPE benchmark provides a **fixed set of candidate** responses for each question, and the reward model's task is to select the correct one. In contrast, evaluating a model directly on MMLU-Pro requires it to generate responses from scratch. Given that the PPE evaluation protocol provides a constrained set of candidates, it is not an appropriate measure for the RL post-training setup.
>
> > … Providing results for intermediate steps could further enhance the robustness of your evaluation.
>
> Thank you for your suggestion on reporting intermediate steps of the experiments. Since we follow your suggestion “comparisons against other … score-based RM”, we add the additional experiments with a save interval of 200 steps, instead of saving every checkpoint. We will include the detailed training logs in our next version due to discussion time limit.
> We believe our extensive experiments on **test-time scaling behaviors** and benchmarks -- including **RewardBench, PandaLM Test, MMLU-Pro, MATH, GPQA, and Arena-Hard** have consistently demonstrated the effectiveness of RRMs.
>
> > I am also interested in the trade-off between compute budget and performance.  … However, inference efficiency is important in practice, since the compute budget for post-training is often limited.
>
> Thank you for your insightful question about concerns of compute budget.
>
> This concern is central to the development of large language models (LLMs). Historically, many groundbreaking models were initially criticized for their computational requirements. However, the rapid advancement of computational power has made these large models a core technique in AI. Our research follows this same principle: our primary focus is to advance the state of the art by effectively **translating increased computational power into greater model intelligence**. The investigation of compute and performance trade-offs is an interesting area for future work, but our primary research contribution is focused on **demonstrating the effectiveness of our approach at scale**.
>
> References:
>
> [1]  How to evaluate reward models for RLHF

---

> ### Comment · Reviewer_DJnD · 2025-08-04
>
> Thank you for your further discussion. While I agree with some of your points, I would like to ask a quick question.
>
> **[W1-1]**
> Thank you for sharing your considerations. I agree that the most important value of a reward model lies in its utility for training, rather than simply serving as a proxy for preference evaluations such as RewardBench. I expect you to reflect this perspective well presented in the next version of your draft.
>
> **[W2-1]**
> Thank you for your detailed reply. However, could you explain further on the significance of the MMLU-Pro improvements (1.66% and 4.27%)? In my opinion, this is the most crucial metric for measuring the effectiveness of the reward model.
>
> **[W2-2]**
> I understand there may have been time constraints in the rebuttal period, but I hope you could consider providing a more robust evaluation in a future revision.
>
> **[W2-3]**
> I appreciate your point that effectiveness is the central focus of the paper. However, some readers or developers may still be concerned about efficiency. I hope this discussion will remain a valuable discussion for the community going forward. I also recognize that the additional experiment you conducted for reviewer mL9a partially addresses my concerns—thank you for that.

---

> ### Author Response · Authors · 2025-08-05
>
> Thanks for your response. We are glad to hear that our responses have addressed your additional concerns.
>
> > … the most important value of a reward model lies in its utility for training, rather than simply serving as a proxy for preference evaluations such as RewardBench …
>
> > However, could you explain further on the significance of the MMLU-Pro improvements (1.66% and 4.27%)? In my opinion, this is the most crucial metric for measuring the effectiveness of the reward model.
>
> Thank you for acknowledging the importance of our experiments on RL post-training/DPO experiments, which are often omitted in concurrent work. Through three experimental setups, we have demonstrated that RRMs can effectively guide post-training with RL or DPO. We believe that the experimental results provide valuable insights into understanding reward reasoning for the community, which is **a contribution, not a weakness**.
>
> Furthermore, in addition to the utility for training, we would like to highlight that reward models also play a substantial role in LLM inference such as reward-guided search [1,2,3], where the results show that test-time compute with reward models can be used to outperform a 14x larger model. Our experiments have demonstrated that best-of-N decoding with RRMs can lead to remarkable gains, **far surpassing “simply serving as a proxy for preference evaluations”.**
> Moreover, since our contribution is to propose a new paradigm— effectively scales with test-time compute in reward modeling. We conduct systematic analysis of the test-time scaling behaviors and patterns, which provides deeper understanding of reward reasoning processes.
>
> Through **systematic analysis on scaling behaviors and patterns**, and extensive experiments on **RewardBench, PandaLM Test, MMLU-Pro, MATH, GPQA, and Arena-Hard**, we believe that our experiments provide solid support of our methods, rather than being focused on improving a single specific benchmark.
>
> > I appreciate your point that effectiveness is the central focus of the paper. However, some readers or developers may still be concerned about efficiency. I hope this discussion will remain a valuable discussion for the community going forward. I also recognize that the additional experiment you conducted for reviewer mL9a partially addresses my concerns—thank you for that.
>
> Thank you for recognizing the value of our research principle. We also believe that inference efficiency / latency is a valuable area for future research, although it is out of the scope of our current work.
>
> References:
>
> [1] Ensembling Large Language Models with Process Reward-Guided Tree Search for Better Complex Reasoning
>
> [2] Scaling LLM Test-Time Compute Optimally can be More Effective than Scaling Model Parameters
>
> [3] Inference-Time Scaling for Diffusion Models beyond Scaling Denoising Steps

---

> ### Author Response · Authors · 2025-08-07
>
> Dear Reviewer DJnD,
>
> Thank you for your time and valuable feedback on our work. We hope that our response and revision have adequately addressed your concerns. As the discussion period nears its end, we would be very grateful if you could reconsider our work and possibly adjust the score accordingly. If you have any additional questions or suggestions, we would be happy to have further discussions.
>
> Best regards,
>
> Authors

---

> > ### Comment · Reviewer_DJnD · 2025-08-08
> >
> > Thank you for your rebuttal and the following discussion. Some of my concerns have been addressed through our exchange. As a result, I have decided to raise my score.

---

> > > ### Author Response · Authors · 2025-08-08
> > > **Thank you for raising your score!**
> > >
> > > Thank you for your response — we’re pleased to hear that your concerns have been addressed and that you’ve decided to raise your score. We sincerely appreciate your thoughtful engagement throughout the discussion.
> > >
> > > As the discussion period is drawing to a close, please don’t hesitate to let us know if you have any remaining questions or concerns. We would be happy to further clarify any aspect of the work.

---

> ### Author Response · Authors · 2025-08-09
> **Thank you for raising your score!**
>
> Thank you for your response — we’re pleased to hear that your concerns have been addressed and that you’ve decided to raise your score. We sincerely appreciate your thoughtful engagement throughout the discussion.
>
> As the discussion period is drawing to a close, please don’t hesitate to let us know if you have any remaining questions or concerns. We would be happy to further clarify any aspect of the work.

---

### Note · Authors · 2025-08-12

We sincerely thank all reviewers for their constructive feedback. Below we summarize the key points.

**Strengths**

1. Novelty & Core Technical Contribution: RRM is novelly designed to perform explicit reasoning before producing final rewards, enabling adaptive compute allocation at inference for effective **parallel/sequential test-time scaling**. Building on proven RL training foundations, this yields a **new and effective paradigm** that drives superior scaling and performance.

2. Comprehensive Evaluation: Best-of-N selection, RL with unlabeled data, reward model-guided DPO, parallel/sequential scaling analyses, and benchmarked across reward evaluation-- including RewardBench, PandaLM Test, MMLU-Pro, MATH, GPQA, and Arena-Hard.

3. Strong Empirical Performance and Scalability: RRM consistently outperforms baselines and other reward models, demonstrating the effectiveness of long reasoning reward modeling. Large-scale experiments (e.g., RRM-32B) further confirm its feasibility and sustained performance gains at scale.

**Key Issues Addressed in the Rebuttal**

1. RL Post-Training with Baseline Comparison: Compared RRM-32B with a strong score-based baseline (Skywork-Reward-Gemma-2-27B); baseline gains degraded in training, while RRM maintained stable improvements and avoided reward hacking.

2. Unlabeled Data on Hard Domains: Conducted test-time RL on challenging mathematical tasks (AIME24) with unlabeled data, showing that RRM-guided training consistently improved performance across domains.

3. Statistical Robustness: Multi-seed runs on RewardBench showed minimal variance, and all reported results are non-cherry-picked.

4. Expanded PPE Evaluation: Added binary preference classification per PPE protocol, confirming RRM’s advantage over strong baselines. (Reviewer 86j6, W2)

5. Advantages over Concurrent Works

    - **Simplicity & Efficiency**: Provides a unified, end-to-end framework that avoids complex, multi-stage pipelines of some concurrent methods (e.g., DeepSeek-GRM), making it easier to implement, train, and maintain.

    - **Performance & Scaling**: Outperforms strong concurrent models (DeepSeek-GRM, RM-R1, J1) on RewardBench and PPE, and continues to benefit from test-time scaling.

We thank all ACs and Reviewers for their valuable input, which helped us substantially improve the clarity and completeness of our paper. We believe that, following the rebuttal process, the quality and impact of our work are sufficient for the community.

---

### Decision · Program_Chairs · 2025-09-17

**Decision:**

Accept (poster)

**Comment:**

This paper introduces Reward Reasoning Models (RRMs), which are trained to generate an explicit chain-of-thought process before producing a final preference score. The primary contribution is a new paradigm for reward modeling where test-time compute can be adaptively scaled to improve accuracy on complex queries, leading to superior performance on reward benchmarks and in downstream applications like reinforcement learning from human feedback (RLHF).

Most reviewers are in agreement, they highlight this work as a novel paradigm for reward modeling, and appreciate comprehensive evaluation across numerous benchmarks and tasks, and strong empirical performance at scale. The initial reviews raised several points for clarification, including the need for stronger baselines in the RLHF experiments, a lack of statistical robustness analysis, and an analysis of the computational overhead. The reviewers were satisfied with the authors' response, which provided significant new experimental results addressing these key points.

For the camera-ready version, the authors are expected to integrate the new experimental results and detailed clarifications from their rebuttal into the main body of the paper. This includes: (1) the new RLHF experiments with a strong score-based reward model baseline, (2) the statistical robustness analysis from multi-seed runs, (3) the expanded evaluation on the PPE benchmark, and (4) the analysis of the additional computational cost of the RRM-based training pipeline.